# Mobilization of endocannabinoids by midbrain dopamine neurons is required for the encoding of reward prediction

Miguel Á. Luján[1,7], Dan P. Covey[1,2,7], Reana Young-Morrison[1], LanYuan Zhang[1], Andrew Kim[1], Fiorella Morgado[1], Sachin Patel [3], Caroline E. Bass [4], Carlos Paladini[5] & Joseph F. Cheer [1,6] ✉

Brain levels of the endocannabinoid 2-arachidonoylglycerol (2-AG) shape motivated behavior and nucleus accumbens (NAc) dopamine release. However, it is not clear whether mobilization of 2-AG specifically from midbrain dopamine neurons is necessary for dopaminergic responses to external stimuli predicting forthcoming reward. Here, we use a viral-genetic strategy to prevent the expression of the 2-AG-synthesizing enzyme diacylglycerol lipase α (DGLα) from ventral tegmental area (VTA) dopamine cells in adult mice. We find that DGLα deletion from VTA dopamine neurons prevents depolarization-induced suppression of excitation (DSE), a form of 2-AG-mediated synaptic plasticity, in dopamine neurons. DGLα deletion also decreases effortful, cue-driven reward-seeking but has no effect on non-cued or low-effort operant tasks and other behaviors. Moreover, dopamine recording in the NAc reveals that deletion of DGLα impairs the transfer of accumbal dopamine signaling from a reward to its earliest predictors. These results demonstrate that 2-AG mobilization from VTA dopamine neurons is a necessary step for the generation of dopamine-based predictive associations that are required to direct and energize reward-oriented behavior.

The brain endocannabinoid (eCB) 2-arachidonoylglycerol (2-AG), detected at concentrations a thousand times higher than those of the other eCB anandamide[1], promotes mesolimbic dopamine release and invigorates motivated behavior[2,3]. It Is generally assumed that activation of the main substrate for 2-AG, the cannabinoid type-1 receptor (CB1R), is responsible for the disinhibition of dopamine neurons located in the ventral tegmental area (VTA)[4]. In contrast, anandamide binds CB1Rs with submaximal potency, thereby blunting accumbal dopamine cell excitability[2,5]. Intriguingly, dopamine neurons are among the few brain cell types with little to no CB1R protein expression[6–8], a surprising phenomenon

considering that CB1Rs are the most abundant G protein-coupled receptor in the brain[9]. A growing body of evidence suggests that CB1Rs (usually coupled to inhibitory $G_{i/o}$ proteins[10]) are strategically positioned throughout the mesocorticolimbic circuitry to fine-tune dopamine neuron excitability[3]. In the VTA, CB1Rs are found in GABAergic and glutamatergic terminals impinging upon dopamine neurons[11]. In their seminal work, Lupica and Riegel (2005)[12] described that eCB release from dopamine cells preferentially activates CB1Rs localized in GABAergic terminals[13], leading to the disinhibition of VTA dopamine cell bodies in a feed-forward loop resulting in a net increase of dopamine outflow in terminal regions[14].

[1]Department of Neurobiology, University of Maryland School of Medicine, Baltimore, MD, USA. [2]Department of Neuroscience, Lovelace Biomedical Research Institute, Albuquerque, NM, USA. [3]Northwestern Center for Psychiatric Neuroscience, Department of Psychiatry and Behavioral Sciences, Northwestern University Feinberg School of Medicine, Chicago, IL, USA. [4]Department of Pharmacology and Toxicology, University at Buffalo, State University of New York, Buffalo, NY, USA. [5]UTSA Neuroscience Institute, University of Texas at San Antonio, San Antonio, TX, USA. [6]Department of Psychiatry, University of Maryland School of Medicine, Baltimore, MD, USA. [7]These authors contributed equally: Miguel Á. Luján, Dan P. Covey. ✉e-mail: jcheer@som.umaryland.edu

While notable progress has been made delineating the neuro-circuitry and electrophysiological properties of the dopamine and eCB systems, less is known about the precise neuropsychological implications emerging from their interaction[15]. It has been theorized, but never unambiguously demonstrated, that 2-AG mobilization from dopamine neurons is a necessary mechanism for the initiation and invigoration of reward-oriented behaviors triggered by conditioned stimuli[16,17]. The primary constraint in providing an explicit demonstration has stemmed from the lack of viral-genetic tools capable of selectively excising –in vivo– distinct molecular components of the eCB machinery deployed by VTA dopamine neurons or their afferent pathways. Advances in mouse mutant lines[18] and viral-genetic recombinant constructs[19] hold great potential to overcome the difficulties inherent to pharmacological approaches. 2-AG is synthesized 'on demand' following increases in $Ca^{2+}$ influx, which activates the catalyzing enzyme diacylglycerol lipase α (DGLα)[20]. Therefore, a selective

experimental strategy arises from the possibility of excising DGLα from midbrain neurons, allowing for the interrogation of 2-AG mobilization and its consequences on downstream dopamine signaling and behavioral functions.

Mesolimbic dopamine neuron projections from the VTA to the nucleus accumbens (NAc) gate the selection and invigoration of appetitive behaviors triggered by outcome-predictive cues[21]. This has been extensively supported by evidence that VTA dopamine neuron firing and NAc dopamine release phasically increase in response to better-than-expected events[22] and, as learning proceeds, the dopamine signal transfers –along with action initiation– to the earliest predictors of forthcoming reward access[23]. This encoding by dopamine signals stands as one of the most replicated phenomena in neuroscience[24]. However, despite its robustness, the underlying molecular mechanisms giving rise to these dynamic signaling changes remain largely unexplored. Pre-existing

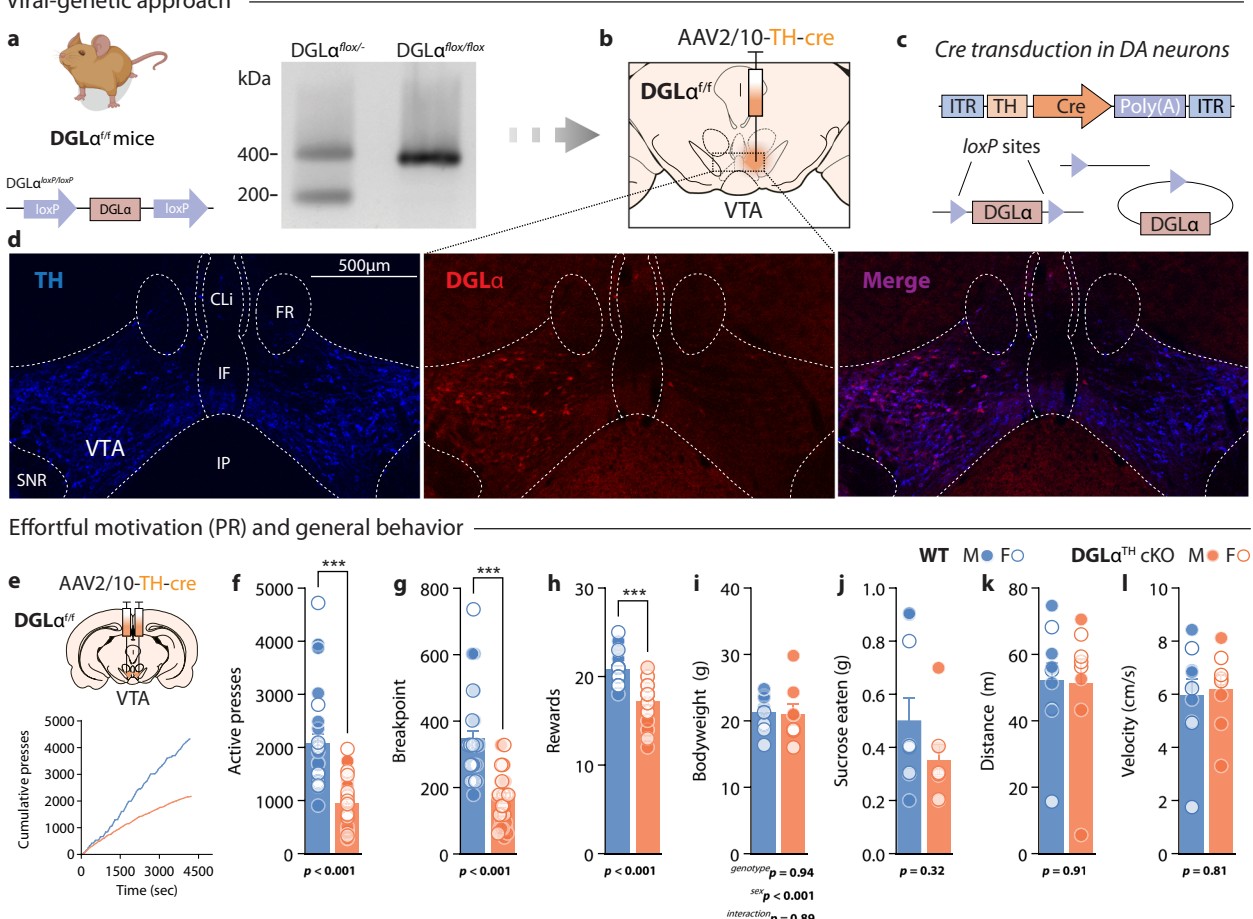

**Fig. 1 | Conditional deletion of DGLα from VTA dopamine neurons in adult animals curtails effortful reward seeking. a** Genomic *loxp* structure surrounding the DGLα (*Dagla*) gene in DGLα[f/f] mice. The right panel shows a PCR genotyping blot of DGLα[f/-] and DGLα[f/f] mice from genomic tail DNA. The 200-bp band corresponds to the WT allele and the 400-bp band corresponds to the gene-floxed allele. **b** AAV2/10-TH-cre construct was bilaterally transduced in the VTA of WT and DGLα[f/f] mice to selectively prevent 2-AG mobilization from dopamine neurons throughout behavioral testing. **c** When the DGLα-floxed sequence recombines with the *Cre* cassettes –exclusively expressed by TH+ cells– conditional excision of the *Dagla* gene from VTA dopamine neurons is achieved. **d** Midbrain immunohisto-chemical confocal images showing co-localization of TH and DGLα in VTA dopamine neurons (left hemisphere untransduced) from a DGLα[TH] cKO animal. DGLα immunoreactivity is prevented after expression of TH-cre in the injected side (right hemisphere). Scale bar, 500 μm. **e** Schematic representation illustrating viral

transduction of VTA dopamine neurons (top panel). Representative cumulative responding during PR for a sucrose pellet reward. Vertical tick marks demarcate reward receipt (bottom panel). **f–h** DGLα deletion from VTA dopamine neurons results in decreased total active lever presses (two-sided Welch's $t_{46.8} = 6.26$, $p = 5 \cdot 10^{-4}$), breaking points (two-sided Mann–Whitney $U = 104$, $p = 1.65 \cdot 10^{-9}$), and rewards earned (two-sided Mann–Whitney $U = 98$, $6.5 \cdot 10^{-10}$) during PR WT, $n = 32$ (17 M,15 F); DGLα[TH] cKO, $n = 32$ (17 M,15 F). **i, j** DGLα[TH] cKO mice displayed similar bodyweight ($t_{16} = 0.18$, $p = 0.85$) and sucrose consumption in a sucrose feeding test (two-sided $t_{16} = 1.07$, $p = 0.32$) [WT, $n = 10$ (6 M, 4 F); DGLα[TH] cKO, $n = 8$ (4 M,4 F)]. **k, l** Total distance traveled (two-sided $t_{16} = 0.11$, $p = 0.91$) and average velocity in an open field test is not affected by the TH-cre expression (two-sided $t_{16} = 0.24$, $p = 0.81$) [WT, $n = 10$ (6 M,4 F); DGLα[TH] cKO, $n = 8$ (4 M,4 F)]. Data are presented as mean ± SEM. Created with BioRender.com.

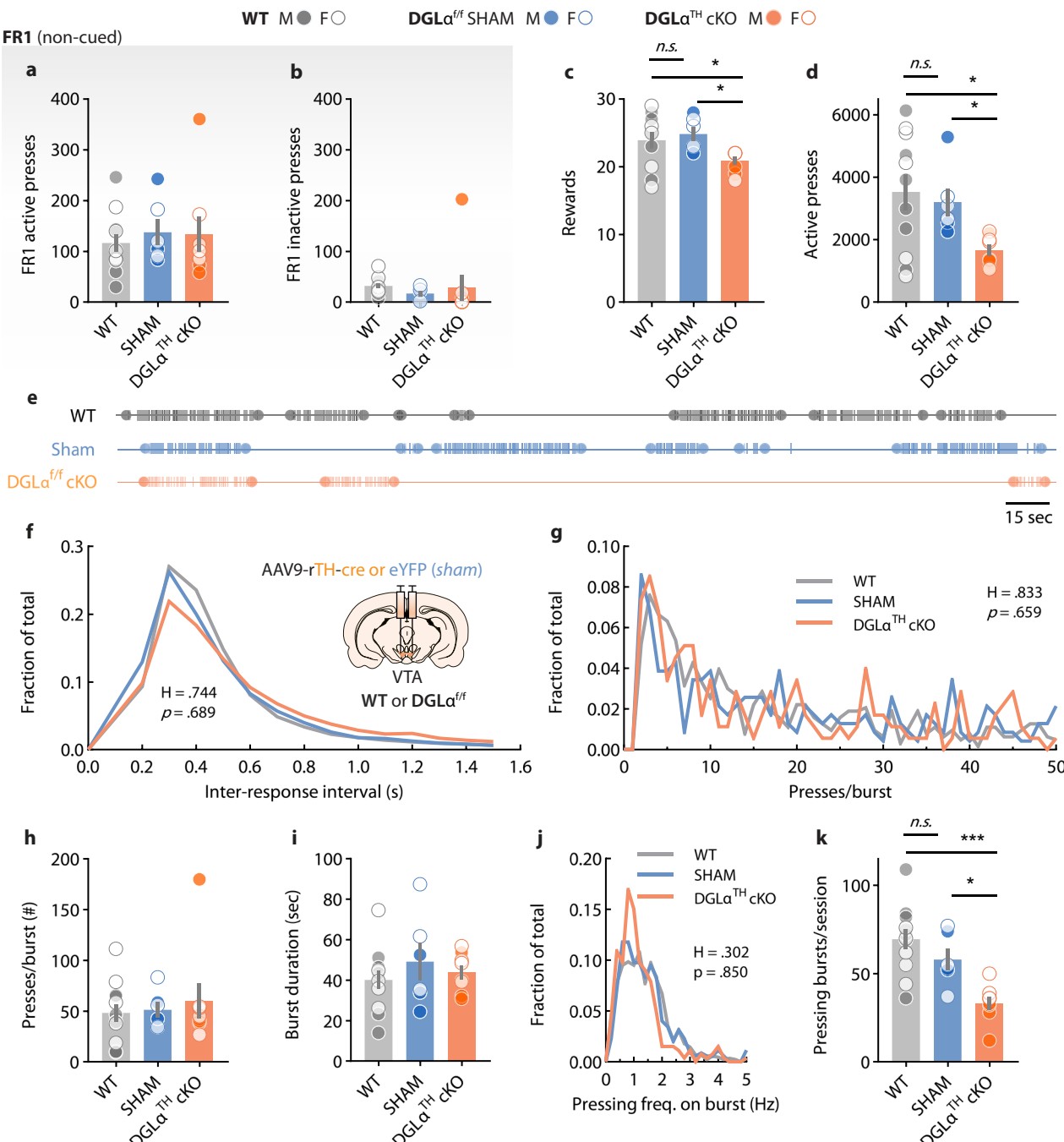

**Fig. 2 | DGLα deletion from VTA dopamine neurons blunts spontaneous task engagement without motor, genotypic or viral confounds. a**, **b** Unchanged non-cued, active and inactive lever pressing behavior (FR1) after DGLα deletion (one-way ANOVAs; $^{active}F_{2,22} = 0.22$, $p = 0.80$; $^{inactive}F_{2,22} = 0.26$, $p = 0.76$) [WT, $n = 12$ (7 M,5 F); DGLα$^{f/f}$ sham, $n = 6$ (3 M,3 F); DGLα$^{TH}$ cKO, $n = 8$ (4 M,4 F)]. **c** During PR, DGLα$^{TH}$ cKO exhibited fewer rewards earned (Kruskal-Wallis; $H_3 = 7.18$, $p = 0.02$) (Dunn's *post hoc* test; WT vs cKO *$p = 0.015$, SHAM vs cKO *$p = 0.028$) [WT, $n = 12$ (7 M,5 F); DGLα$^{f/f}$ sham, $n = 6$ (3 M,3 F); DGLα$^{TH}$ cKO, $n = 8$ (4 M,4 F)] and (**d**) total active lever presses (Welch's ANOVA; $F_{2,10.89} = 5.54$, $p = 0.01$) (Dunnett's *post hoc* test; WT vs cKO *$p = 0.023$, SHAM vs cKO *$p = 0.039$) [WT, $n = 11$ (6 M, 5 F); DGLα$^{f/f}$ sham, $n = 6$ (3 M,3 F); DGLα$^{TH}$ cKO, $n = 8$ (4 M,4 F)]. **e** Representative lever-pressing timestamps. Lines correspond to individual lever press. Circles indicate the first and last lever press of each 'burst'. **f** Frequency distribution of inter-response intervals

during PR testing (Kruskal-Wallis; $H = 0.74$, $p = 0.68$). **g** Number of lever presses per burst distributed equally among groups (Kruskal–Wallis; $H = 0.83$, $p = 0.65$). **h** Lever presses on each responding burst did not vary across groups (one-way ANOVA; $F_{2,22} = 0.27$, $p = 0.76$) [WT, $n = 12$ (7 M, 5 F); DGLα$^{f/f}$ sham, $n = 6$ (3 M,3 F); DGLα$^{TH}$ cKO, $n = 8$ (4 M,4 F)]. **i** Average duration of lever pressing burst was not affected (one-way ANOVA; $F_{2,22} = 0.65$, $p = 0.52$) [WT, $n = 12$ (7 M,5 F); DGLα$^{f/f}$ sham, $n = 6$ (3 M,3 F); DGLα$^{TH}$ cKO, $n = 8$ (4 M,4 F)]. **j** All three groups displayed similar distributions of high- and low-frequency lever pressing bursts during PR (Kruskal–Wallis; $H = 0.30$, $p = 0.85$). **k** DGLα$^{TH}$ cKO exhibited a reduced number of lever pressing bursts (one-way ANOVA; $F_{2,23} = 12.20$, $p = 0.002$) (Holm-Šídák; WT vs cKO **$p = 1.6 \cdot 10^{-4}$, SHAM vs cKO *$p = 0.018$) [WT, $n = 11$ (6 M,5 F); DGLα$^{f/f}$ sham, $n = 6$ (3 M,3 F); DGLα$^{TH}$ cKO, $n = 8$ (4 M,4 F)]. Data are presented as mean ± SEM.

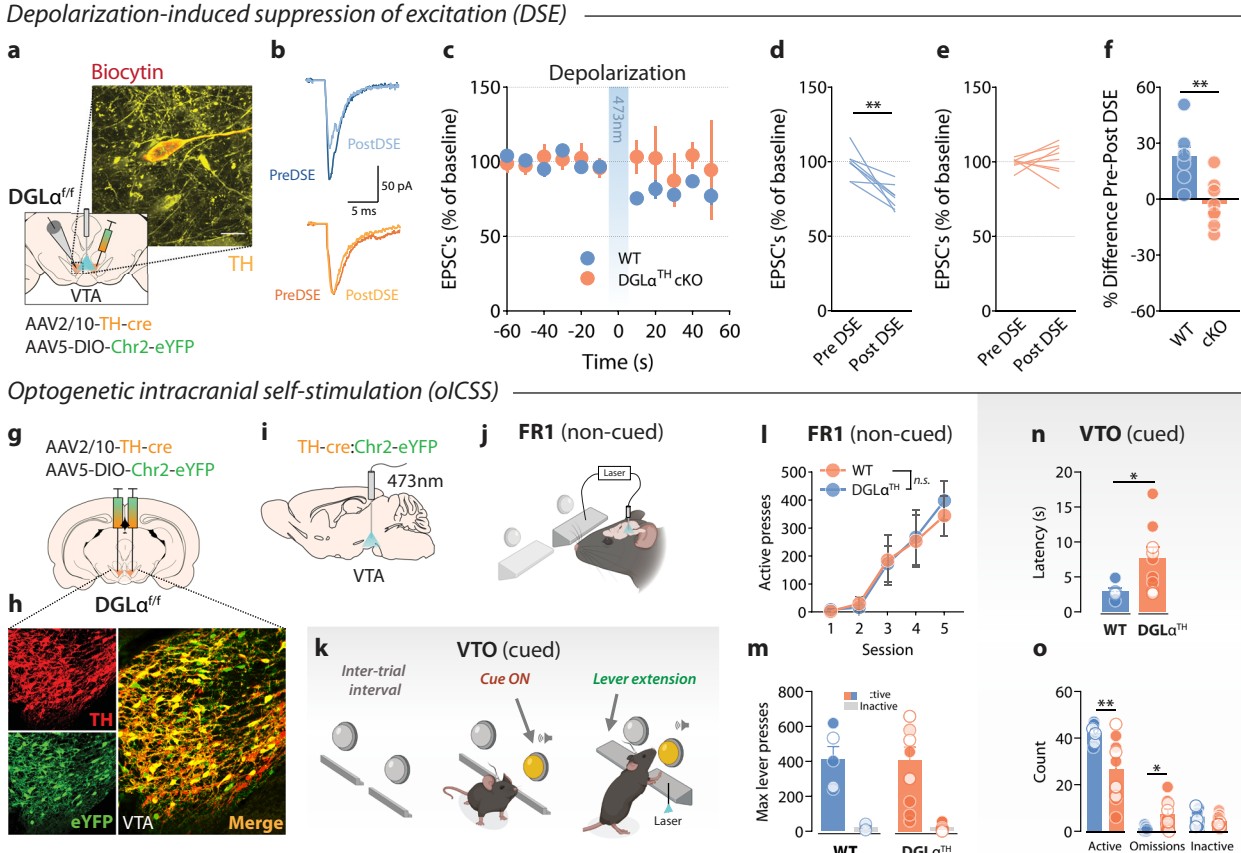

**Fig. 3 | DSE and cued reward seeking require 2-AG mobilization from VTA dopamine neurons. a** DSE set-up and confocal image of a midbrain section in which one TH+ cell was patched and filled with biocytin. Scale bar = 20 μm. **b** Representative traces from dopamine cells showing electrically-evoked EPSCs before and after DSE. **c** Time course of electrically-evoked EPSCs before and after optically-evoked DSE (WT, $n = 8$; DGLα$^{TH}$ cKO, $n = 8$). **d–f** Averaged percentual changes in EPSCs amplitudes before and after DSE (paired $^{WT}t_7 = 5$, **$p = 0.001$; paired $^{DGLα}t_7 = 0.57$, $p = 0.58$; $^{ratio}t_{14} = 4.07$, **$p = 0.001$; all two-sided) (WT, $n = 8$; DGLα$^{TH}$ cKO, $n = 8$). **g** Schematic illustration of TH-cre viral transduction in VTA dopamine neurons. **h** Confocal images (20x magnification) showing immunostaining for anti-TH, eYFP (DIO-Chr2) and its merge in the VTA. **i** An optical fiber was placed in the VTA to allow for closed-loop lever press-laser stimulation of Chr2-expressing TH+ neurons. **j** Schematic illustration of closed-loop optostimulation of VTA dopamine neurons during non-cued FR1. **k** Alternatively, mice underwent VTO

testing. On each VTO trial, response latency is determined as the time elapsed between lever extension and lever press. **l** Unchanged FR1 oICSS responding across five 30-min operant sessions following conditional deletion of DGLα (two-way RM ANOVA; $^{genotype}F_{1,13} = 0.01$, $p = 0.90$; $^{session}F_{4,52} = 13.54$, $p = 6.1·10^{-5}$; $^{session x}$ $^{genotype}F_{4,52} = 0.10$, $p = 0.97$). **m** Maximum active and inactive lever-pressing behavior throughout FR1 (two-sided $^{active}t_{13} = 0.02$, $p = 0.98$; two-sided $^{inactive}t_{13} = 0.36$, $p = 0.71$) (WT, $n = 5$; DGLα$^{TH}$ cKO, $n = 10$). **n** In presence of exteroceptive guiding cues (VTO), rapid conditioned approach is lost after DGLα deletion (two-sided Welch's $t_{9.23} = 2.92$, *$p = 0.017$) (WT, $n = 5$; DGLα$^{TH}$ cKO, $n = 10$). **o** Curtailed reward-paired cue processing results in reduced active lever pressing (two-sided Welch's $^{active}t_{10.3} = 3.13$, **$p = 0.01$) and increased omission errors (two-sided Welch's $^{omissions}t_{8.8} = 2.84$, *$p = 0.02$) despite similar inactive lever presses (two-sided $^{inactive}t_{13} = 1.14$, $p = 0.27$) (WT, $n = 5$; DGLα$^{TH}$ cKO, $n = 10$). Data are presented as mean ± SEM. Created with BioRender.com.

pharmacological evidence indicates that elevations of 2-AG tone[25], or activation of CB1R[2], boost execution of effort and dopaminergic encoding of reward-predictive cues. In light of this evidence, we hypothesize that mobilization of 2-AG is a requisite step for VTA dopamine neurons to progressively sculpt phasic dopamine release events to the earliest predictors of forthcoming reward. Subsequently, we postulate that midbrain dopamine neuron 2-AG mobilization promotes rapid cue-evoked reward seeking behaviors and effortful behavioral engagement.

Here, we sought to isolate the functional relevance of the 2-AG-synthesizing enzyme (DGLα) in VTA dopamine neurons in behaving mice using a viral-genetic approach with negligible effects on motor function and non-cued operant responding but striking implications for the acquisition and invigoration of conditioned reward seeking and accompanying dopamine signals. To achieve this, a *cre* recombinase vector driven under the promoter of tyrosine hydroxylase (TH) –a dopamine neuron marker– was injected into the VTA of DGLα$^{f/f}$ adult mice, therefore preventing the ability of this neuronal population to mobilize 2-AG in behaving animals.

## Results

### Depletion of 2-AG from VTA dopamine neurons curtails effortful motivation

To assess the contribution of 2-AG mobilization from ventral tegmental area (VTA) dopamine neurons in motivated behavior, we expressed *cre* recombinase under a tyrosine hydroxylase (TH) promoter in the VTA of wild-type (WT) and DGLα$^{f/f}$ mice (Fig. 1a-c). Figure 1d shows a representative midbrain section of a DGLα$^{f/f}$ animal following unilateral transduction of the AAV2/10-TH-cre construct. Co-expression of TH and DGLα was evident in the non-transduced side, confirming prior reports that dopamine neurons contain the enzymatic machinery required for 2-AG production[26,27]. In the transduced side, DGLα expression was excised from TH+ cells (Fig. 1d, middle panel). Moreover, we co-transduced a *cre*-dependent eYFP construct (AAV5-DIO-eYFP) in the VTA of WT mice to confirm the specific expression of TH-driven *cre* recombinase in VTA dopamine neurons (Supplementary Fig. 1). Prior to the initiation of the main series of experiments, WT and DGLα$^{f/f}$ *sham* mice were subjected to a battery of behavioral tests to screen for non-specific genotypic differences. Our

results indicate that floxing the *Dagla* gene did not affect locomotion in an open field test or anxiety-like behaviors in the elevated plus maze (Supplementary Fig. 2), as previously documented[18]. After expressing AAV2/10-TH-cre bilaterally in the VTA of WT and DGLα^f/f mice (hereafter referred to as WT and DGLα^TH cKO, respectively) (Fig. 1e), male and female animals were trained to lever press for sucrose rewards on a progressive ratio (PR) schedule of reinforcement, where response requirement grew exponentially to obtain a single sucrose pellet. Once this occurred, both levers retracted, and the house and cue lights dimmed for 20 s. Four PR sessions were conducted and data from the last session –once the lever pressing behavior was acquired– is depicted in Fig. 1. Under these conditions, depletion of 2-AG production from VTA dopamine neurons potently curtailed effortful reward seeking (Fig. 1f-h) (n = 32/genotype). The addition of sex as a factor in these metrics discarded sex-dependent effects on active presses (2-way ANOVA; $^{sex}F_{1,62} = 0.08$, $p = 0.76$; $^{sex \ x \ group}F_{1,62} = 0.99$, $p = 0.32$) breaking points ($^{sex}F_{1,62} < 0.01$, $p = 0.99$; $^{sex \ x \ group}F_{1,62} = 0.92$, $p = 0.33$) or rewards earned ($^{sex}F_{1,62} = 0.62$, $p = 0.43$; $^{sex \ x \ group}F_{1,62} = 2.35$, $p = 0.13$). Additional metrics were obtained following PR testing to eliminate non-specific confounds related to bodyweight (Fig. 1i), free-feeding sucrose consumption (Fig. 1j), or open-field locomotor function (Fig. 1k, l) following TH-cre transduction in WT and DGLα^TH cKO mice.

Beyond effortful motivation, VTA dopamine pathways are closely linked to motor output and basic learning functions[28]. To discard non-specific effects of viral vector infusion and dopamine cell DGLα knock-down, we delivered an independently developed TH-cre construct, carried by a different serotype (AAV9-rTH-cre), into the VTA of male and female WT and DGLα^f/f mice (Supplementary Fig. 3). To rule out genotype-related spurious effects during operant responding, a DGLα^f/f *sham* group, spared from AAV9-rTH-cre transduction, was included (Fig. 2). This time, animals were first trained to seek sucrose pellets on a continuous fixed-ratio 1 reinforcement schedule during which no conditioned cues were present (non-cued FR1). WT, DGLα^f/f *sham*, and DGLα^TH cKO exhibited similar active and inactive FR1 lever pressing performance (Fig. 2a, b), suggesting no impairments in low-effort reward seeking due to genotypic- or viral delivery-related causes. Furthermore, these results suggest that DGLα function in VTA dopamine neurons does not impact low-effort, goal-directed behavior. Next, mice progressed to PR testing. DGLα^TH cKO mice exhibited reduced levels of effortful motivation compared to WT and DGLα^f/f *sham* in terms of active lever pressing or rewards earned (Fig. 2c, d). We assessed potential motor confounds during the PR session by analyzing the distribution of spontaneous lever presses clustered in response 'bursts'. Figure 2e illustrates individual lever presses displayed by three representative WT, DGLα^f/f *sham* and DGLα^TH cKO mice for a 6-min interval. Reward seeking behavior (that is, lever presses) displayed characteristic spontaneously-clustered lever pressing 'bursts'. Should motor deficits arise from our manipulations, then it would be reasonable to expect significant changes in how lever presses clustered within each engagement event: i.e., reduced response density within each 'burst' (fewer presses per 'burst)'. In contrast, changes in how lever-pressing behavior is hierarchically structured during the task (i.e., total number of 'bursting' events) should be independent of potential motor impairments, but rather reflect the motivation of the animal to engage in reward-oriented behaviors. The distribution of inter-response intervals during the PR session revealed that the TH-cre viral injection, DGLα^f/f genotype, or its combination (DGLα^TH cKO) were not associated with slower rates of responding (Fig. 2f). Consistent with the lack of motor deficits, WT, DGLα^f/f *sham* and DGLα^TH cKO showed no differences in the number of lever presses per 'burst' (Fig. 2g, h). In all three groups, lever pressing bursts were of similar duration (Fig. 2i) and, crucially, no deviations in lever press density were observed at either low- or high-responding frequency bouts (Fig. 2j). Finally, we found that the number of lever-pressing 'bursts' was significantly lower in DGLα^TH cKO mice compared to their WT and

DGLα^f/f *sham* counterparts (Fig. 2k), supporting lower PR performance consistent with changes in the structure of reward-directed behavior throughout the task (number of engagements) but not in lever-pressing motor execution.

## Downregulated VTA dopamine cell plasticity and cued responding in DGLα^TH cKO mice

Depolarization-induced suppression of excitation/inhibition (DSE/I) is a hallmark form of 2-AG-mediated synaptic plasticity[29,30]. Upon depolarization, dopamine neurons retrogradely release 2-AG in a DGLα-dependent manner, activating CB1R located at presynaptic glutamate and GABA terminals[31]. This phenomenon is hypothesized to result in a loss of feedforward excitation/inhibition of dopamine neuron bursting activity[32], ultimately shaping phasic dopamine release at terminal regions such as the nucleus accumbens (NAc)[33,34]. Here, we examined whether conditional inducible deletion of DGLα from dopamine neurons resulted in the loss of DSE. Specifically, WT and DGLα^f/f mice were bilaterally transduced with AAV2/10-TH-cre in addition to AAV5-DIO-Chr2-eYFP, hence allowing us to optogenetically depolarize dopamine neurons (Fig. 3a). Single VTA dopamine neurons in midbrain hemi-slices were recorded in the whole-cell configuration while holding the cells in voltage-clamp at +40 mV. We observed a decrease in excitatory post-synaptic currents (EPSCs) recorded from dopamine neurons following a depolarizing optogenetic stimulation train (473 nm at 6–7 mW, 10 s) in WT but not DGLα^TH cKO samples (Fig. 3b–f). To further test the CB1R-dependency of this mechanism, we bath-applied the CB1R antagonist AM251 during the depolarizing optostimulation, which completely prevented the appearance of DSE in WT slices (Supplementary Fig. 4). These findings support the viability of the viral-genetic approach at the cellular level, indicating that DGLα^TH cKO VTA dopamine neurons cannot communicate retrogradely via the eCB 2-AG with incoming CB1R-expressing afferents. Moreover, the abolition of DSE reveals a fundamental involvement of DGLα – and the resulting "on demand" release of 2-AG onto presynaptic CB1Rs – in the orchestration of VTA dopamine cell plasticity.

To better characterize downstream behavioral consequences of deficient 2-AG production by dopamine neurons, we expressed AAV2/10-TH-cre and AAV5-DIO-Chr2-eYFP in the VTA of WT and DGLα^TH cKO mice to simultaneously prevent 2-AG production and enable optogenetic intracranial self-stimulation (oICSS) of dopamine neurons[35] (Fig. 3g, i). Figure 3h shows co-localization of TH and eYFP (ChR2 reporter) immunostaining in a VTA coronal section. We also implanted three WT mice with a carbon fiber microelectrode aimed at the NAc, which allowed us to validate the oICSS procedure via fast-scan cyclic voltammetry (FSCV) (Supplementary Fig. 5). Optical stimulation of the VTA (473 nm at 6–7 mW, 1 s) produced frequency-dependent (10–50 Hz) increases in NAc dopamine release (Supplementary Fig. 5). *Closed-loop* photoactivation of VTA dopamine cells served to fine-tune the stimulation parameters (473 nm at 6–7 mW, 30 Hz, 1 s) to better resemble spontaneous NAc dopamine release events in response to rewarding stimuli. Next, WT and DGLα^TH cKO animals were trained to lever-press for VTA laser self-stimulation throughout five FR1 and one variable time-out (VTO) oICSS sessions. These operant tasks were chosen to selectively deconvolve the role of outcome-predictive cues in the motivational phenotype associated with DGLα deletion from dopamine neurons. During FR1 reinforcement (30-min sessions), actuation of the laser-coupled lever resulted in non-cued optostimulation of VTA dopamine neurons (Fig. 3j). However, during VTO reinforcement mice had to lever-press in response to a light-tone compound cue indicating forthcoming access (5 s) to a reward-associated lever (Fig. 3k). Animals had 60 s to respond following lever extension; failure to press resulted in an omission error and the start of a new VTO period (10–90 s). Latency to press upon lever presentation following cue onset was used as a proxy of the cue's association with

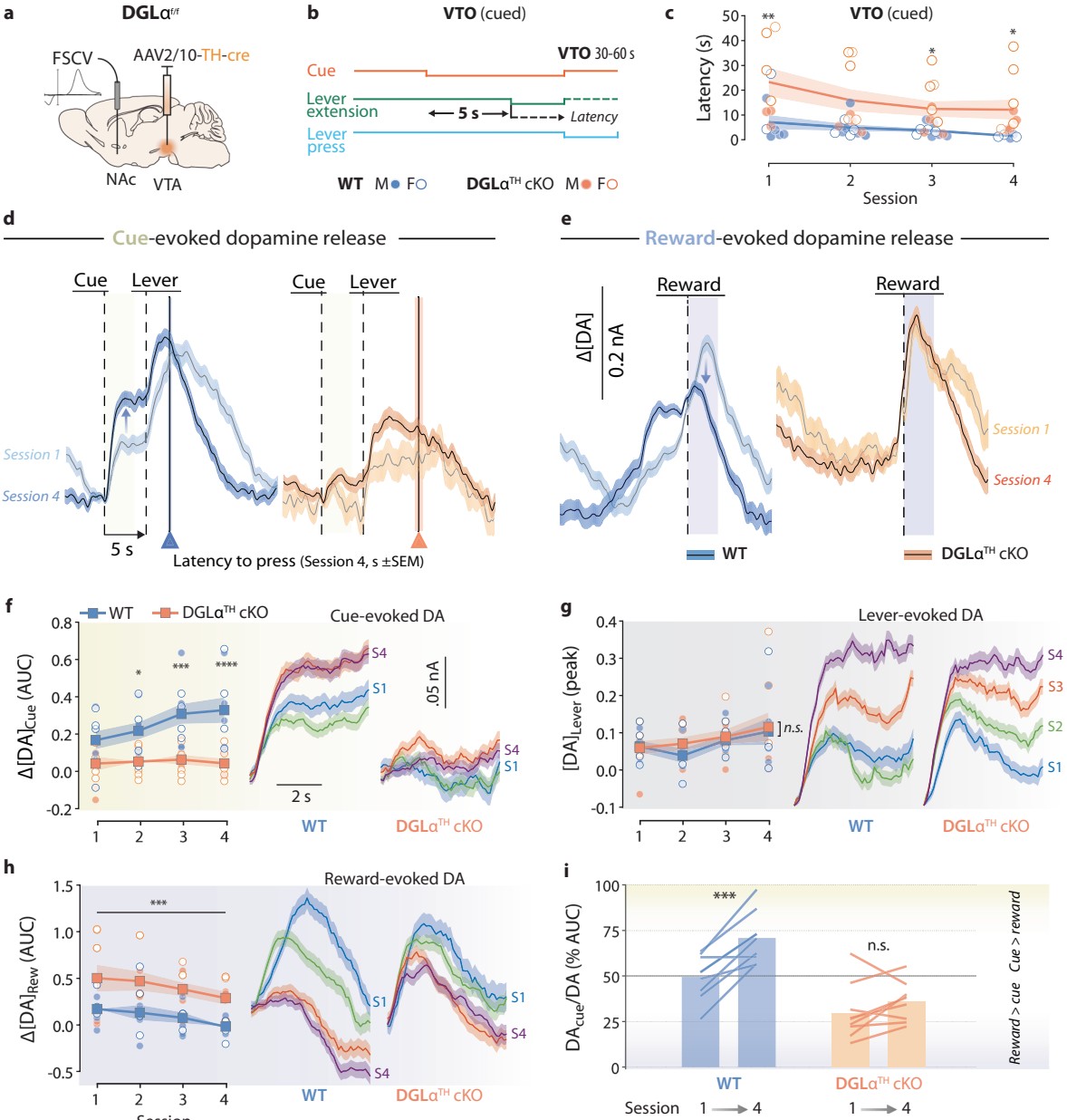

**Fig. 4 | 2-AG mobilization is necessary for the transfer of dopamine encoding from rewards to their earliest predictors. a** In vivo FSCV recordings of NAc phasic dopamine release in WT and DGLα^TH cKO mice. **b** Schematic representation of VTO trial structure. In this experiment, the compound cue is interpreted as the earliest predictor of reward. Lever extension is considered a proximal cue. **c** DGLα^TH cKO mice showed longer response latency to press across four VTO sessions (two-way RM ANOVA; $^{genotype}F_{1,16} = 8.41$, $p = 0.01$; $^{session}F_{3,48} = 6.03$, $p = 0.01$) (WT vs. DGLα^TH cKO, same session, Holm-Šídák $^*p = 0.04$, $^{**}p = 0.004$). **d, e** Representative NAc dopamine concentration change time-locked to cue (**d**) and reward delivery (**e**) onset over time. Traces from the first and last VTO sessions are shown. The vertical black line represents the group's mean latency to lever press (s ± SEM, colored area) on the last VTO session. **f** Diminished NAc dopamine release upon distal cue presentation in DGLα^TH cKO mice (two-way RM ANOVA; $^{genotype}F_{1,16} = 17.67$, $p = 6.7 \cdot 10^{-4}$;

$^{session}F_{3,48} = 3.14$, $p = 0.03$) (WT vs. DGLα^TH cKO, same session, $^*p = 0.02$, $^{***}p = 6.4 \cdot 10^{-4}$, $^{****}p = 9.4 \cdot 10^{-5}$). **g** Unchanged dopamine response to proximal cue (lever extension) presentation (two-way RM ANOVA; $^{genotype}F_{1,16} = 0.23$, $p = 0.63$; $^{session}F_{3,48} = 4.02$, $p = 0.01$; $^{session \times genotype}F_{3,48} = 0.34$, $p = 0.78$). **h** Increased NAc dopamine release at the time of reward delivery in DGLα^TH cKO mice (two-way RM ANOVA; $^{genotype}F_{1,16} = 12.44$, $p = 0.002$; $^{session}F_{3,48} = 4.39$, $p = 0.03$) (WT vs cKO, $^{***}p = 0.002$). **i** Compared to WT, NAc dopamine release from DGLα^TH cKO animals does not progressively track cue onset over reward delivery as training progresses (two-way RM ANOVA; $^{genotype}F_{1,16} = 23.58$, $p = 1.7 \cdot 10^{-4}$; $^{session}F_{1,16} = 26.50$, $p = 9.7 \cdot 10^{-5}$; $^{session \times genotype}F_{1,16} = 7.93$, $p = 0.01$) (session 1 vs. 4, same genotype, Holm-Šídák $^{***}p = 7.5 \cdot 10^{-5}$) (WT, $n = 9$, 5 M, 4 F; DGLα^TH cKO, $n = 9$, 4 M, 5 F). Group-averaged FSCV traces of each recording session are shown in the right panels. Data are presented as mean ± SEM.

forthcoming reinforcement. Our results indicate that responding to the laser-paired lever only differed between groups during cued reinforcement (VTO) but not during non-cued reward seeking (FR1) (Fig. 3l–o). Considering the contrasting consequences of our genetic manipulation, this evidence suggests that VTA dopamine cells mobilize 2-AG via DGLα to invigorate reward seeking behavior

in scenarios requiring the internalization of exteroceptive guiding cues (such as reinforcement during VTO) and high motivational demand (e.g., PR). Furthermore, the continuous FR1 results indicate that DGLα deletion does not affect dopamine neuron function, per se, because direct depolarization was similarly reinforcing in both groups.

## Deficient dopaminergic encoding of reward's earliest predictors in DGLα$^{TH}$ cKO animals

Invigoration of appetitive behaviors critically relies on accumbal dopaminergic encoding of outcome-predictive cues[28,36], which progressively transfers from reward itself to the onset of the earliest predictive cues as training proceeds[37,38]. However, the precise molecular mechanisms that allow dopamine neurons to transfer responding from rewards to its earliest predictors are largely unknown. Prior pharmacological evidence suggests that eCB neurotransmission controls stimulus-encoding by dopamine neurons via 2-AG mobilization onto CB1Rs expressed on presynaptic terminals[25]. This idea is also in alignment with our abovementioned behavioral results. Hence, to formally test this hypothesis, animals expressing TH-cre in the VTA (Fig. 4a) were implanted with FSCV probes in the NAc and underwent a VTO task (Fig. 4b) to simultaneously assess learning-related dopamine dynamics and cue-elicited sucrose reward seeking[39]. Animals were tested during four consecutive sessions, allowing us to track how phasic dopamine transients transferred to the earliest predictors of reward. As previously seen with self-optostimulation, DGLα$^{TH}$ cKO mice exhibited higher response latencies (Fig. 4c). This decrease in response vigor resulted in a higher number of omission errors and fewer sucrose rewards earned, despite similar levels of inactive lever pressing (Supplementary Fig. 6). Moreover, this effect was not caused by non-specific influences of the DGLα$^{f/f}$ genotype, as behavioral differences in the VTO task were absent in DGLα$^{f/f}$ *sham* mice (Supplementary Fig. 6).

To assess dopaminergic encoding of this task, we focused on dopamine release events time-locked to the delivery of sucrose pellets (reward), as well as the onset of proximal (lever extension) and distal predictors (compound light + tone cue). Representative FSCV color plots and in vivo NAc dopamine concentration traces can be seen in Supplementary Fig. 6. Greater cue-evoked dopamine signals reflected a dopamine-based attribution of salience to reward-predictive cues[36] as large cue-evoked NAc dopamine transients preceded lower response latencies (Supplementary Fig. 6). Longer response latencies, instead, were associated with larger reward-evoked transients (Supplementary Fig. 6). Figure 4d, e shows that dopamine release differentially tracked the onset of distal (cue) and proximal (lever) predictors and reward delivery in both WT and DGLα$^{TH}$ cKO. Firstly, genetic ablation of VTA 2-AG production from dopamine neurons attenuated cue-evoked NAc dopamine responses to the earliest predictor of forthcoming reward (Fig. 4f). Secondly, lever-evoked NAc dopamine concentration changes did not vary as a function of our manipulation (Fig. 4g), discarding gross alterations in dopamine release function after genetic ablation of DGLα and supporting a more nuanced role for 2-AG in the appraisal and exploitation of distal, but not proximal, outcome-predictive cues. Thirdly, DGLα$^{TH}$ cKO displayed greater dopaminergic responses following sucrose pellet delivery compared to WT mice (Fig. 4h).

More importantly, learning-dependent dynamics of dopamine release in DGLα$^{TH}$ cKO mice significantly deviated from the patterns expected from the conceptual framework of dopamine signals, and those observed in WT mice. NAc phasic dopamine transients in response to the distal cue progressively increased across sessions in WT animals (Fig. 4f). This canonical neural response was completely abolished in DGLα$^{TH}$ cKO mice (Fig. 4f). The evolution of lever-evoked dopamine transients followed a similar pattern, but this pattern of encoding was not influenced by genetic deletion of DGLα from VTA dopamine neurons (Fig. 4h), indicating unaltered dopaminergic attribution of salience to proximal cues in DGLα$^{TH}$ cKO mice and discarding uncontrolled learning deficits. Finally, blunted 2-AG midbrain mobilization resulted in persistent, monolithic phasic dopamine transients upon reward delivery in DGLα$^{TH}$ cKO mice, in contrast to the progressive decrease in reward-evoked dopamine release by WT animals (Fig. 4h). Putatively deficient dopamine encoding dynamics in DGLα$^{TH}$ cKO mice was further characterized by comparing each animal's ratio of dopamine release allocated to reward or its earliest predictor (cue) in early *versus* late VTO sessions (Fig. 4i). Our results reveal a remarkable disparity between groups, illustrating the inability of DGLα$^{TH}$ cKO dopamine neurons to transfer NAc release events from reward to its earliest predictor across training. Overall, these observations are consistent with a deficient process of salience attribution to distal, but not proximal, cues predicting forthcoming access to reward, consequently accompanied by an impairment in conditioned responding in 2-AG-deficient mice.

## Discussion

By modulating limbic-motor striatal regions[40], phasic dopamine release allows cues to rapidly optimize behavioral resources, hence promoting fast approach toward rewards[41]. Here, we demonstrate cell type-specific 2-AG signaling is required for distal outcome-predictive cues to effectively convey information about forthcoming rewards. This conclusion is supported by the compromised invigoration of cue-directed behavior in high-demand scenarios (short latency requirements) as well as the lack of dynamic range observed for accumbal dopamine encoding of reward-predictive distal cues and rewards. In addition, these effects were accompanied by a notable decrease in effortful motivation. Importantly, experimentally induced dopamine cell-specific impairment of 2-AG mobilization was not linked to gross motor output alterations, basic associative learning deficits or reduced operant reward seeking in scenarios not requiring the exploitation of predictive cues or high effort. These studies support the critical role of 2-AG signaling by VTA dopamine neurons to sculpt accumbal release patterns consistent with encoding and appraisal of distal outcome-predictive cues.

Our data is consistent with the notion that VTA dopamine neurons communicate retrogradely with CB1R-expressing GABAergic afferences via DGLα-synthesized 2-AG, providing a disinhibitory mechanism[34]. That is, when dopamine neurons burst at high-frequency, elevation in intracellular Ca$^{2+}$ levels lead to DGLα catabolic activity[42,43]. Consequently, synthesis and retrograde release of 2-AG activates CB1R located on GABAergic terminals, thus procuring the suppression of GABA release[30]. This dopamine neuron disinhibition results in an increase in dopamine terminal release[44]. Thus, this empirical evidence supports a role for midbrain eCB signaling as a potential substrate sustaining the backpropagation of motivational information throughout the mesolimbic circuitry, closing the loop between *top-down* afferents to the VTA and *bottom-up* dopamine release[45].

In our experimental conditions, DGLα ablation negated the ability of VTA dopamine cells to backpropagate information about its depolarization events, which resulted in the absence of DSE. DSE/I is the most commonly observed form eCB-based plasticity that allows dopamine neurons to accommodate the influence of excitatory (DSE) or inhibitory (DSI) inputs to their burst firing events[46]. Of note, the loss of DSE reported in DGLα$^{TH}$ cKO animals is not thought to directly participate in the behavioral and dopaminergic effects reported here. Blunting CB1R-induced inhibition of glutamatergic influences (DSE) should result in the facilitation of dopamine cell function and its related functions, in opposition to the phenotype here unveiled. Instead, we postulate that a loss of DSI was responsible for the observed effects, as it is consistent with a disinhibition of GABAergic input onto VTA dopamine neurons. Here, to demonstrate the effectiveness of our viral-genetic strategy, we used DSE, therefore assessing the effectiveness of eCB-mediated synaptic plasticity in recordings of adult VTA dopamine neurons[47].

eCBs are amongst the most ubiquitous signaling molecules in the brain and, consequently, a large volume of work has described its many functions in emotional regulation, motor function and memory[48]. We thoroughly examined phenotypic differences to rule

out the influence of extraneous effects of our viral-genetic model that could affect dopaminergic, behavioral or learning performance. Our data indicates that DGLα deletion from VTA dopamine cells did not change free-feeding, non-operant consumption of sucrose. It neither altered exploratory nor anxiety-like responses. Moreover, we conducted an in-task characterization of motor skills presumably involved in lever pressing, which would introduce a notable confound in the present findings. Nonetheless, our results discarded that down-regulation of 2-AG mobilization from dopamine cells impaired the motor execution of lever pressing, as we did not detect changes in how individual responses spontaneously clustered in high-frequency responding events (or lever pressings 'bursts'). Instead, DGLα$^{TH}$ cKO engaged fewer times in reward-directed behaviors (number of 'burst' events), a metric independent of lever press execution. Next, we demonstrated that no gross alterations in dopamine release and basic learning functions were present in DGLα$^{TH}$ cKO mice. First, we proved that VTA dopamine neurons deprived of 2-AG synthesis can still sustain self-optostimulation and act as a primary reinforcer. Second, we revealed that DGLα$^{TH}$ cKO also exhibited dopamine predictive encoding in the VTO task, as lever extension (a proximal predictor of reward availability) triggered normal NAc dopamine elevations known to precede action initiation toward appetitive stimuli[39,49,50]. The increased lever-evoked dopamine signal across sessions may be indicative of faster action initiation toward the lever (i.e., decreased latency to press), which was evident in both groups, albeit at different rates. However, this signal was not transferred to the antecedent cue following deletion of DGLa in VTA dopamine neurons. This finding also discards perceptual impairments related to the sensory characteristics of the specific compound light+sound distal cue here employed. We also observed that deletion of DGLα in VTA dopamine neurons increased reward-evoked NAc dopamine release during VTO. This is congruent with an underutilization of outcome-predictive cues preceding the reward's availability. According to formal learning theories of dopamine signals, unpredicted rewards trigger greater release of NAc dopamine compared to otherwise predicted outcomes[51,52]. Interestingly, goal-seeking behavior in low-demand and non-cued scenarios (FR1) remained unaffected following genetic DGLα removal from VTA dopamine neurons. This finding supports that exploitation of cues becomes increasingly relevant as behavioral and temporal resources become scarce[53,54], hence maximizing an organism's evolutionary fitness to the environment when access to rewards is compromised. However, the question remains unresolved as to whether the mobilization of midbrain 2-AG is distinctly involved in both effortful motivation (PR) and cue-driven responding (VTO), or if these phenomena stem from a shared underlying cause. In our previous research utilizing an adapted PR schedule, we observed that the encoding of NAc dopamine in response to distal cues varies as a function of the anticipated effort needed to attain a reward[25]. Notably, cues indicating a high-effort trial trigger dopamine responses of reduced magnitude compared to cues indicating low effort[25]. Given the compromised dopaminergic encoding of distal cues exhibited by DGLα$^{TH}$ cKO mice, deficient dopamine encoding of such distal cues during PR responding (in this case, house-light onset) could explain the diminished execution of effort in absence of 2-AG mobilization. To ascertain the interdependence of these phenomena, future experiments should involve monitoring dopamine release through a PR chain schedule of reinforcement, wherein distal cues denote distinct effort requirements. Finally, no spurious effects of the DGLα$^{f/f}$ genotype were evident, as the disruption of effortful motivation and rapid conditioned responding was absent in DGLα$^{f/f}$ *sham* mice.

In light of these findings, we propose that production of the endogenous CB1R ligand 2-AG by VTA dopamine neurons is a necessary mechanism for this cellular population to function as the canonical neural substrate of reward prediction during reward seeking.

## Methods

All experimental procedures conformed to the National Institute of Health *Guide for the Care and Use of Laboratory Animals*. Ethical approval was granted by the Institutional Animal Use and Care Committee at the University of Maryland, Baltimore (IACUC protocol #00000054).

### Subjects

For all experiments, we used either wild-type C57BL6/J or DGLα$^{f/f}$ on a C57BL6/J background mice (3–6 months old, female and male subjects). DGLα$^{f/f}$ mice were obtained from Vanderbilt University[55]. DGLα$^{f/f}$ mice were maintained by homozygote x homozygote breeding. Genotypes were determined by PCR of genomic tail DNA using the following primers (5'–3'): TGAGCCAGAGACATTTGCTG, CTGGTGAGGCCAAGTTTGTT and GGGACAGAAAACCACTTGGA. Animals were housed in a temperature- and humidity-controlled room (24 °C and 40–50% humidity, respectively) and maintained on a 12 h light/dark cycle (07:00–19:00 h). All experiments were conducted in the light cycle. The number of mice for each experiment are indicated in the respective figure legends. Sex of the animals is reported throughout the figures and its legends, with open circles representing female subjects and closed circles referring to male mice. Sex was added as an additional factor in all the parametric analyses performed. When no sex effect was detected, males and females were collapsed in the same group and analyses proceed without including sex as a biological variable. A significant sex difference was only observed for body weight but no interaction with our experimental manipulations was found (Fig. 1i; two-way ANOVA: $^{sex}F_{1,14} = 18.7$, $p < 0.001$; $^{genotype}F_{1,14} = 0.005$, $p = 0.94$; $^{interaction}F_{1,14} = 0.89$, $p = 0.35$).

### Histology

Mice underwent isoflurane anesthesia (5%) and transcardial perfusion with a 4% paraformaldehyde (PFA) solution in a 0.1 M sodium phosphate buffer (PB) at pH 7.4. Following perfusion, brains were post-fixed at 4 °C in PFA overnight. Brain sections, (40 μm thick) were obtained using a vibratome (Leica). For the immunohistochemistry of tyrosine TH DGLα, coronal sections were immersed in a PB solution containing 0.2% Triton X-100 (Sigma-Aldrich) and 3% normal donkey serum (Jackson 017-000-121) for a duration of 30 min. Subsequently, brain sections were incubated with primary antibodies (mouse monoclonal 1:1000 anti-TH; ImmunoStar, Catalog# 22941; and guinea pig 1:500 anti-DGLα, gifted by Dr. Ken Mackie, Indiana University) overnight. Secondary antibodies (1:000 donkey anti-mouse Alexa 647, Jackson 715-605-151 and 1:1000 donkey anti-guinea pig Alexa 488, Jackson 706-545-148), were applied for 2 h, followed by staining with 4',6-diamidino-2-phenylindole (DAPI, 1:50,000). Sections were then mounted on slides for visualization of TH, DGLα, and eYFP. Imaging was performed using a confocal microscope (Olympus Fluoview, Tokyo, Japan). Animals lacking viral expression in the target region were excluded from the data analysis.

### RNAscope in situ hybridization (ISH)

RNAscope ISH was employed to ascertain the cell type-specific expression of Cre recombinase mRNA within dopamine TH-expressing cells. Following deep anesthesia induction, mice brains were promptly removed and frozen at −80 °C for coronal sectioning (14 μm thick) in a Leica microtome. The resulting brain slices were then mounted on glass slides (Fisher Scientific). Mounted samples were kept at −80 °C until ISH assays were conducted. *Cre*, *Th* and *Dagla* mRNAs cellular distributions in the VTA were detected using the following RNAscope probes (obtained from ACDbio, Newark, CA, US): Mm-Th-C2 (*Th*; cat. no. 317621-C2), Mm-Dagla-C3 (*Dagla*; cat. no. 478821-C3) and Mm-CRE-C1 (*Cre*; cat. no. 312281). The RNAscope mRNA assays were performed following the manufacturer's protocols. Stained slides were covered with DAPI mounting medium

(Fluoroshield ab104139; Abcam) and scanned into digital images with an Olympus Fluoview confocal microscope at 20× magnification.

## Slice electrophysiology

DGLα[f/f] mice were transduced with AAV2/10-TH-cre in the VTA and left undisturbed in their home-cage for 3 weeks. On test day, mice were decapitated, and their brains were rapidly removed and transferred to an oxygenated (95% $O_2$/5%$CO_2$) ice-cold solution containing: 93 mM NMDG, 2.5 mM KCl, 1.2 mM $NaH_2PO_4$, 30 mM $NaHCO_3$, 20 mM HEPES, 25 mM Glucose, 5.6 mM Ascorbic acid, 3 mM Sodium pyruvate, 10 mM $MgCl_2$, 0.5 mM $CaCl_2$. Horizontal slices containing VTA (220 μm) were transferred to a holding chamber filled with oxygenated solution containing: 109 mM NaCl, 4.5 mM KCl, 1.2 mM $NaH_2PO_4$, 35 mM $NaHCO_3$, 20 mM HEPES, 11 mM Glucose, 0.4 mM Ascorbic acid, 1 mM $MgCl_2$, 2.5 mM $CaCl_2$. Slices were initially incubated at 35 °C for a duration of 10–12 min. Subsequently, they were transferred to room temperature, where they remained until the experiments began. The slices were then moved to a recording chamber and submerged in constantly-flowing (2 ml/min) oxygenated artificial cerebrospinal fluid (aCSF) (32–34 °C). The aCSF composition consisted of 3 mM KCl, 26 mM $NaHCO_3$, 2.4 mM $CaCl_2$, 126 mM NaCl, 11 mM Glucose, 1.2 mM $NaH_2PO_4$, and 1.5 mM $MgCl_2$. Visualization of the slices was achieved using a differential interference contrast (DIC) optics upright micro-scope (Olympus, BX51WI). We recorded lateral VTA neurons located anterior to the third cranial nerve and medial to the terminal nucleus of the accessory optic track.

Whole-cell voltage-clamp recordings were obtained with an Axo-patch 200B amplifier (Molecular Devices). Recording pipettes with a resistance of 3–5 MΩ were filled with an internal solution containing: 2 mM NaCl, 140 mM K-gluconate, 1.5 mM $MgCl_2$, 10 mM Tris-phos-phocreatine, 10 mM HEPES, 0.3 mM Na-GTP, 4 mM Mg-ATP, 0.1 mM EGTA. The pH of the internal solution was adjusted to 7.2, and the osmolarity was maintained at 290 mOsm. Picrotoxin (50 mM) was added to the aCSF for recordings to isolate excitatory transmission. EPSCs were evoked using a train of five-stimuli (100 μs, 1 mA) delivered at 50 Hz every 30 s with bipolar tungsten stimulating electrodes with tip separation 100–200 μm. The amplitudes of EPSCs were calculated by taking a 1 ms window around the peak of the EPSC and comparing this with the 5 ms window immediately before the stimulation artifact. The depolarizing pulse used to evoke depolarization-induced sup-pression of excitation (DSE) was achieved by optostimulation of ChR2-expressing TH+ neurons (473 nm; 10 s; 6–7 mW, 30 Hz). The magni-tude of DSE was measured as percentage of the mean amplitude of consecutive EPSCs after depolarization (acquired between 10 and 50 s after the end of the pulse) relative to that of five EPSCs before the depolarization. Stimulation protocols were generated and signals acquired using the WinLTP program. In an additional experiment (Supplementary Fig. 4), control EPSCs were recorded for 60 s before and after the CB1R blocker AM251 (2 μM) was bath-applied. Each slice received only a single drug exposure. Data are presented as the change in percent from control traces.

## Surgical procedures

Mice were anesthetized with isoflurane in O2 (4% induction and 1% maintenance, 2 L/min). Then, to induce a conditional DGLα knock-out in VTA dopamine neurons, 300 nl/side of AAV2/10-TH-iCre (synthesized by and obtained from Dr. Caroline Bass, University of Buffalo)[19] were injected bilaterally into the VTA (−3.3 AP, +0.5 ML, −4.0 DV, mm relative to bregma) of DGLα[f/f] mice. In a separate set of animals (Supplementary Fig. 1), AAV2/10-TH-iCre was co-infused with AAV9-Ef1a-DIO-eYFP (Addgene viral prep #27056) in a 500 nl/ side 1:1 mixture. For the ex vivo electrophysiological experiments (Fig. 1q), AAV2/10-TH-iCre was co-infused with AAV5-EF1a-DIO-hChR2(h134r)-eYFP (University of North Carolina, Vector Core) in a 500 nl 1:1 mixture. Alternatively, we injected AAV9-rTH-PI-Cre-SV40

(300 nl/side, Addgene viral prep #107788, packaged by the UMB vector core), or AAV9-Ef1a-DIO-eYFP as a control, to replicate the main behavioral findings with an independent viral construct and serotype (Supplementary Fig. 3). Viral injections (0.1 μl/min) used graduated pipettes (Drummond Scientific Company), broken back to a tip diameter of ~20 μm. For the oICSS experiments (Fig. 2), an optical fiber (105 μm core diameter, 0.22 NA, Thorlabs, NJ) was then implanted unilaterally above the injection site at −3.8 mm DV. For FSCV recordings, a chronic voltammetry electrode was then also implanted ipsilateral to the optical fiber in the NAc core ( +1.2 AP, +1.1 ML, −3.7 DV, mm relative to bregma) and an Ag/AgCl reference electrode in the contralateral superficial cortex, as described previously[41,56,57]. All components were permanently affixed with dental cement (Metabond, Parkell, Inc). Mice were allowed 4 weeks to recover from surgery and allow viral expression.

## Behavior

There were four main series of behavioral experiments, each one corresponding to each main figure. Mice underwent either sucrose PR (Fig. 1), sucrose FR1/PR (Fig. 2), oICSS FR1/VTO (Fig. 3), or multiple sucrose VTO (Fig. 4) testing. Details about each specific behavioral task are described below.

**Operant training schedules.** Lever pressing shaping and training varied across experiments. (1) For the experiments reported in Fig. 1, WT and DGLα[TH] cKO mice underwent four PR sessions. Data from the last session is shown. (2) WT, DGLα[f/f] *sham*, and DGLα[TH] cKO mice, appearing in Fig. 2, first underwent three FR1 sessions (30-min). Data from the last session –once lever pressing behavior was acquired– is shown (Fig. 2a, b). After that, animals progressed to PR testing. Four sessions were conducted and data from the last session is shown in panels Fig. 2c–k. (3) For oICSS experiments, WT and DGLα[TH] cKO animals (Fig. 3) were first trained in VTO responding and then pro-gressed to the FR1 sessions (Fig. 3l–o). (4) For the FSCV/VTO experi-ments, highlighted in Fig. 4, animals were exclusively tested on VTO responding, as described below. All operant data was collected using Med-PC® IV.

**Apparatus.** In all cases, mice were tested in operant chambers (21.6 × 17.6 × 14 cm; Med Associates) housed within sound-attenuating enclosures. Each chamber was equipped with two retractable levers (located 2 cm above the floor), one LED stimulus light and a 2.5 kHz tone-generating speaker located above each lever (4.6 cm above the lever). A houselight and a white-noise speaker (80 dB, masking noise background) were located on the opposite wall. For FR1, PR and VTO experiments, an external food magazine was placed within the box that delivered sucrose pellets (14 mg; Bio-Serv) to a dispenser centrally located between the two levers. Two weeks after surgical procedures, mice were mildly food deprived (85–90% of starting body weight), receiving standard laboratory mouse chow daily in addition to food rewards earned during task performance.

**Fixed Ratio 1 (FR1).** When FR1 training was required (see "Operant training schedules" subsection), mice were trained under FR1 schedule of reinforcement with a 10 s timeout in 30-min daily sessions. During FR1 training, animals had continued access to the active and inactive levers (except for the timeout period) and each active lever press was rewarded with a sucrose pellet. Pellet acquisition did not have any other consequences. Responses on the inactive lever were recorded but had no programmed consequences. For the experiment shown in Fig. 2, animals were trained on FR1 until stable responding was estab-lished (<15% variation in response rate across 3 consecutive sessions). For the oICSS cohort (Fig. 3l), FR1 training elapsed 2 more days, totaling five consecutive sessions.

**Progressive ratio (PR).** A progressive ratio schedule of appetitive reinforcement was used to estimate the effort mice were willing to expend for a sucrose pellet reward. On each successive trial, the response requirement (lever presses) needed to obtain a reward scaled near-logarithmically, as determined by the function 'response requirement = (5 x $e^{(0.2 \times reward\ number)}$ - 5)', after rounding to the nearest integer. The response ratio of the first sixteen trials was: 1, 2, 4, 6, 9, 12, 15, 20, 25, 32, 40, 50, 62, 77, 95 and 118. The last response requirement attained, also known as breakpoint, was recorded and used to infer the inherent motivation for the reward. At the beginning of each session, both active and inactive levers were extended, accompanied by the illumination of the cue light placed on top of the active lever and the house light. Actuation of the inactive lever had no consequences, but inactive responses were recorded. Upon response requirement completion, a pellet was delivered, and, for 10 s, levers retracted, lights were turned off and a 2.5 kHz tone was emitted. After the end of this 10 s time-out period, a new trial began. The PR session ended whenever no reward could be obtained within 20 min. Four PR sessions were conducted and data from the last session, –once the lever pressing behavior was acquired– is depicted.

**Variable time-out (VTO).** First, mice underwent two acclimation sessions in which non-cued sucrose pellet rewards were non-contingently delivered following a VTO schedule with inter-trial intervals (30 trials total) ranging from 30–60 s (average 45 s). Then, mice were transferred to an operant VTO (30–60 s) operant schedule of reinforcement, similar to that used in prior work[2,39,41]. In this task, the cue light was illuminated and a 2.5 kHz tone was played 5 s prior to lever extension. Presses on the active lever immediately delivered a sucrose pellet if the response requirement was met within 60 s following lever extension. If the response requirement was not met within this time frame, both levers retracted, the cue light was turned off, and the trial was counted as an omission. Presses on the inactive lever were recorded but had no programmed consequences. A total of 80 trials were presented every session (four total). FSCV recordings were performed every session. For control purposes, a group of DGLα[f/f] *sham* mice were run along WT and DGLα[TH] cKO animals (shown in Supplementary Fig. 6) but no FSCV recordings were carried out.

**Elevated plus maze (EPM).** WT, DGLα[f/f] *sham* mice were tested for anxiety-like behaviors using a procedure similar to one previously reported[58]. Briefly, The EPM test took place in a black maze that was elevated 30 cm off the ground and lit from above. Each mouse was placed in the center of the maze and allowed to explore for 5 min. The total time spent in the open arms were recorded using the EthoVisionXT 17 video-imaging system. The total number of entries (where all four paws were placed in the arm) was also counted.

**Sucrose consumption.** To discard changes in innate sucrose palatability, a sucrose feeding test was performed under *ad libitum* feeding conditions. WT and DGLα[TH] cKO (Fig. 1j) mice were placed in an open-field arena (1 m × 1 m) and allowed free access to sucrose pellets for 30 min.

**Open field test (OFT).** To assess overall motor function, an open-field locomotor test was performed with WT, DGLα[f/f] *sham* (Supplementary Fig. 2), or DGLα[TH] cKO' (Fig. 1k, l) in a 1 × 1 m arena for 1 h. Behavior was recorded with a digital video camera, positioned overhead. Data were analyzed using EthoVisionXT 17 video-imaging system.

**oICSS**

Four weeks after surgery, WT and DGLα[f/f] mice were placed into operant chambers for oICSS training. Animals were trained on reinforcement schedules; each active lever response led to delivery

of a train of light stimulation. Light was delivered by a diode-pumped solid-state laser (473 nm, 150 mW) coupled to 62.5 mm core, 0.22 NA optical fiber (Thor Labs). Light output was -10–20 mW at the tip of the ferrule. In all cases, laser stimulation consisted of 4 ms pulses during 1-s at 30 Hz. Presses on one lever produced immediate laser stimulation accompanied by a 1-s illumination of the cue light and activation of the 2.5 kHz tone placed above the lever, while presses on the other lever (inactive) or presses on the active lever during an ongoing stimulation (non-reinforced) were collected but had no programmed consequences. A separate group of WT mice (Supplementary Fig. 5) that were also implanted with FSCV recording electrodes were initially trained for a minimum of 3 sessions on an FR1 schedule and then, on a separate session, received non-contingent 1-s trains of light stimulation at varying frequencies (10–50 Hz, 473 nm, 6–7 mW) to determine the relationship between frequency stimulation and accumbal dopamine release.

**In vivo FSCV**

As in prior studies[25], voltammetry was employed to monitor dopamine concentration variations. To do so, a triangular waveform (−0.4 to +1.3 V at 400 V/s) was applied (10 Hz) to carbon fiber microelectrodes aimed at the NAc of WT and cKO mice. Redox reactions around the carbon fiber tip were detected as fluctuations in faradaic current from which concentration changes of the electroactive analyte were derived. Thus, all voltammetric measurements are reported in current units (nA). Data was collected using LabView 2020. Using principal component regression (PCR), we statistically derived the dopaminergic component of the voltammetric measurements obtained during cue onset and the delivery of the reward (5-s window)[59]. Lever-evoked dopamine levels were calculated over the total time elapsed between lever extension and pellet delivery on a trial-by-trial basis. A visual representation of the time intervals used for quantification measurements is depicted in Supplementary Fig. 6e, g. Baseline was determined during the 0.5 s that preceded the compound light+tone. Training sets were created using non-contingent optogenetically evoked DA signals (Supplementary Fig. 5) and a standard set of five basic pH shift voltammograms.

**Statistics and reproducibility**

Parametric behavioral and voltammetric measures were analyzed using a one- or two-way repeated measures (RM) ANOVA, or unpaired *t* tests and Holm–Šídák *post hoc* test was used to correct for multiple comparisons when appropriate. For all parametric analyses, Welch's corrections were applied to all the cases wherein violations of homoscedasticity were detected. If so, Welch's-corrected analyses were followed by Dunnett's post hoc test. When analyzing ordinal variables (rewards/breaking points on PR task), the non-parametric Mann–Whitney statistic was employed in unifactorial designs and the Kruskal Wallis test in multifactorial cases, followed by Dunn's *post hoc* tests. Significance was set at *p* < 0.05 and all tests were two-tailed. Statistical analyses were performed in Prism 9.5 (GraphPad), Demon Analysis Software, Matlab 2020a and SigmaPlot 14.5. Statistical details and sample size for each experiment are presented in figure legends. No statistical methods were used to pre-determine sample sizes. Confirmation of viral expression and optic fiber/electrode implantation in each mouse was done through histology. Similar expression and/or localizations (Figs. 1a, d, and 3a, h) were repeatedly observed in minimally 3 different independent samples. The experimenters were blinded to the animals' genotype during the experimental procedures.

**Reporting summary**

Further information on research design is available in the Nature Portfolio Reporting Summary linked to this article.

## Data availability

The data that support the findings of this study have been deposited on a Figshare repository under accession code doi.org/https://doi.org/10.6084/m9.figshare.24299074[60]. Source data are provided with this paper.

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

## Acknowledgements

This research was supported by the National Institute on Drug Abuse (R01 DA022340 to J.F.C., R00 DA047432, K99 DA047432 and F32 DA041827 to D.P.C.).

## Author contributions

Conceptualization: D.P.C., J.F.C. and M.Á.L. Methodology: D.P.C., J.F.C., M.Á.L., S.P., C.E.B. and C.P. Formal analyses: D.P.C., J.F.C. and M.Á.L. Investigation: D.P.C., M.Á.L., R.Y.M., F.M., LY.Z., A.K. and C.P. Data curation: D.P.C. and M.Á.L. Writing – original draft: D.P.C., J.F.C. and M.Á.L. Writing – review and editing: D.P.C., J.F.C. and M.Á.L. Visualization: D.P.C. and M.Á.L. Supervision: D.P.C. and J.F.C. Project administration: D.P.C. and J.F.C. Funding acquisition: D.P.C. and J.F.C.

## Competing interests

The authors declare no competing interests.
