## [Peer Review File · Nature Communications]

Mobilization of endocannabinoids by midbrain dopamine neurons is required for the encoding of reward predictionREVIEWER COMMENTS

Reviewer #1 (Remarks to the Author):

The study by Luján et al provides evidence that endogenous cannabinoid signaling in dopamine neurons contributes to invigoration of behavior by dopamine and encoding of reward-predictive cues by dopamine transients. The study has many strengths, including technical expertise and converging approaches, inclusion of males and females, the use of conditional knock-outs, inclusion of df and other statistical information, and consideration of a range of behaviors when phenotyping mice. The manuscript is well written, which is no small feat for such a short paper.

My major concern/question is about the control groups used to compare the conditional knockout. Specifically, most of the experiments compare DGLa-f/f mice with the conditional KO to WT C57BL6/J mice rather than to DGLa-f/f mice without the conditional KO (or SHAM). The authors first cross DAT-cre mice with WT C57, DGLa-f/-, and DGLa-f/f, and find that the resulting lines do not differ in a variety of behaviors, but do differ in motor coordination (roto-rod). It is possible that reduced motor coordination could impact the speed of operant responding – how do the inter-press intervals compare across groups? The remainder of the study used DGLa-f/f mice with a conditional KO using AAV2/10-TH-cre (or other AAV) in the VTA. The one test I found of the DGLa-f/f SHAM mice in effortful responding was in Supplemental Figure 4B – these data showed that the number of presses in a PR task was similar in the DGLa-f/f SHAM versus the C57 WT, and the conditional KO reduced presses. Adding a comparison of the speed of lever press behavior (inter-press interval) across the three groups would inform the interpretation of the PR data – can the rats press quickly, or do they have a motor deficit (e.g. motor incoordination)? The remainder of the comparisons compared C57 WT to the DGLa-f/f conditional KO. In the studies assessing the use of predictive cues to guide behavior (VTO), C57BL6/J mice are used as a control rather than DGLa-f/f SHAM mice. But in order to interpret the behavioral deficit, we need to know how the DGLa-f/f SHAM perform in the VTO task. This is important because the VTO task includes higher order conditioning that is more than simply effortful (such as a FR1 or PR task) – it should be clear that the DGLa-f/f SHAM mice perform as well as C57 mice in order to interpret the deficits observed in the conditional KO mice.

This concern carries into Figure 3, which shows that the dopamine release associated with 4 sessions of VTO differs between C57 WT to the DGLa-f/f conditional KO. The two lines of mice exhibited different dopamine release amplitude to the cue and reward from the first session, which persisted across 4 sessions. As differences in cue-evoked versus reward-evoked dopamine have been reported in various mouse and rat strains (e.g. with sign-tracking and goal-tracking), it is important to know how much of the difference in dopamine from the C57 mice is due to the conditional KO and how much is due to the DGLa-f/f mice. Note that there are two cues in this task – the light/tone cue and the lever extension, and in the representative trace the DGLa-f/f conditional KO mouse shows dopamine release to the lever extension, but not to the light/tone cue. Separating the dopamine response to these cues may uncover starker effects of the conditional KO and help to interpret the precise difference in dopamine signaling. Similarly, the behavioral response to the lever is described as latency to approach/press, but what is the behavioral response to the light/tone in the two genotypes?

Additional comments:

While the paper states that both males and females were used, no indication of the proportion of each sex is provided. Please add #M and #F per group in the figure legends.

The paper states that mice between 3-6 months old were used. As that's a broad range, and older rodents can exhibit lower dopamine release and take longer to learn tasks, please show that mouse age is generally matched across groups/genotypes.

It would be helpful to the reader to refer to the conditional KO mice as DGLa-f/f cKO rather than just DGLa-f/f (which could be SHAM, non-manipulated, or cKO).

Please add to the methods or figure caption how the change in dopamine concentration was measured – what was the baseline for each event? Did the cue response AUC include the lever extension?

Reviewer #2 (Remarks to the Author):

In the study by Lujan et al, the authors examined the role of endogenous 2-AG mobilization in the ventral tegmental area (VTA) on behavior and dopamine release in the nucleus accumbens (NAc). To address this question, they used a viral-genetic approach to reduce the expression of diacylglycerol lipase a (DGLa), which is critical for 2-AG synthesis. Given that many studies on cannabinoid function are based on agonist administration, this approach and the resulting findings are highly significant to the field as they can uncover the endogenous function of endocannabinoid signaling on neurotransmission and behavior. Overall, this study indicates that the dopamine response to visual cues and the corresponding behavioral response to those cues are regulated by endogenous 2-AG signaling in the VTA. A particular strength of the manuscript is the robust behavioral effects of knocking down DGL expression in the VTA on cued (but not uncued) tasks. While there is significant enthusiasm for this study, there are some notable issues that need to be addressed that are listed below. Many of the concerns can likely be addressed if this paper were reformatted for a full-sized article versus the current brief communication format.

Major concerns (in order of appearance):

1. Please provide the behavioral performance of the animals on the FR1 training prior to the PR sessions. Also, clarify the session duration of the FR1 training. Specifically, one needs to know if decreasing DGL expression in the VTA led to differences in the operant responding during the FR1 phase, as this could confound the interpretation of the data from the subsequent PR sessions. Please also provide further details on how many PR sessions the animals underwent.
2. The behavioral experiments provided in Extended Data Fig 1 using the DAT^{cre} x DGL^{f/f} mice are not an appropriate control for the experiments in the rest of the study, which instead used a viral approach to express Cre in TH expressing neurons in the VTA of adult DGL^{f/f} mice. The solely genetic approach used in Extended Data Fig 1 could lead to developmental compensation. The behavioral characterization experiments examining if there are gross impairments should use the same viral-genetic approach that is used in the main figures. Additionally, the rationale for these chosen behavioral assays in Extended Data Fig 1 wasn't evident. Specifically, what is the purpose of the Y-maze assay? And provided the results are consistent when using the viral-genetic approach, how should one interpret the

finding that knocking down DGL expression in the VTA improves motor skills on the rotarod assay? I am also not sure these experiments are needed given the data shown in Fig 1J-L. More important control experiments are outlined in Point 7 below.

3. Please report the sex for the subjects in each figure. Also please show the individual data points in the summary bar graphs throughout the study.

4. The absence of DSE in the DGL^{af/f} mice is strong electrophysiological evidence that the viral-genetic approach is working as intended. However, it is not clear that this electrophysiological effect is contributing to the observed effects on dopamine release and behavior. This again may be an issue with the brief communication format. When reformatting this paper, it would be recommended to highlight this data as functional evidence that the manipulations are working (versus saying this is a plausible mechanism underlying the behavioral effects).

5. There are some questions regarding the analysis of the dopamine signals. It isn't clear why the cue response is calculated over a 3s window when the cue is presented for 5s. I am assuming that the 3s cue window is designed to match the 3s reward window for the analysis in Fig. 3G. But the analysis in Fig. 3G is also not intuitive regarding how it was calculated. Specifically, what is contributing to the denominator? If it is just the sum of the calculated Cue response (over 3 s) and the calculated reward response (over 3 s), then this means there is a 2 s window during the cue that is not being analyzed? Further details are needed regarding this analysis. In short, I would recommend performing the cue analysis over the full 5s of the cue.

6. The dopamine analysis is focused on the presentation of the cue lights and following the reward delivery. However, the average dopamine traces illustrate the dopamine response also increase when the lever is presented, though this is not quantified. The lever-evoked response is important to examine as this response seems to increase over training in the mice that lack DGL in the VTA (Fig. 3H). Conceptually, the lever presentation could be seen as another reward-predictive cue, so one should be cautious on how to interpret these results. In short, it is important to quantify the lever-evoked dopamine response. I would recommend quantifying the average dopamine between the lever presentation and the delivery of the food reward, which would have to be calculated on a trial-by-trial basis.

7. One of the most intriguing findings was the absence of the dopamine response to the cue light turning on in the VTO task in the mice lacking DGL in the VTA (Fig. 3H). However, as

discussed above, these mice seem to exhibit an increase in the dopamine response to the lever presentation, which is the most proximal reward-predictive stimulus. This opens the question if the absence of DGL in the VTA is impairing the detection of visual stimuli or alternatively is impairing the detection of distal reward-predictive cues. To address this question one could train mice on a Pavlovian conditioning task using visual stimuli or audio stimuli (in separate groups of animals). If there is a light above the food tray, one could illuminate that light for 5s before the reward delivery. Alternatively, one could illuminate both of the cue lights (no lever extension) for 5s before the reward delivery. For the audio Pavlovian task, one could present a tone for the 5s before the reward delivery. These stimuli should presumably promote goal tracking behavior since no sign tracking behavior can be exhibited. By comparing the anticipatory head entries during the cues between WT and DGLf/f mice in the audio and visual Pavlovian tasks, one can determine if the deficit in DGLf/f mice is to visual stimuli (if impairments are seen only in the visual Pavlovian task) or to distal reward-predictive cues (if no impairments are seen in both Pavlovian tasks) or is evident only in operant tasks (if impairments are seen in both Pavlovian tasks). I hesitate to ask for new experiments, but in this case I feel that it is quite important given the stated conclusions of the study. However, if there is no effect of training on the lever-evoked dopamine response in the DGLf/f mice (see Point 6), the proposed Pavlovian experiments would be unnecessary as the data would then indicate that training did not impact the dopamine response to either the distal cue (light) or the proximal cue (lever).

8. Histology figures for voltammetry electrodes and optogenetic fibers are missing from the manuscript.

Text issues:

1. Details on the sucrose consumption and OFT refer to Q175 mice, which presumably was data contained in a different study.

2. Figure 1D legend has the orientation backwards regarding the left/right of where the injections were given.

3. Details on the sucrose feeding test are not provided (Fig. 1J). Only those for the 24hr sucrose preference test are provided (Extended Fig. 1). Or is the data in Fig. 1J reflecting the

sucrose preference test but then converted to grams?

4. Text refers to AM experiments as Ext Fig 4, but it is Ext Fig 5. The dotted lines in the figure are confusing. Were those cells that underwent 2 rounds of DSE induction? If so, those should be removed from the analysis.

5. This should be written and formatted as a full manuscript.

Answers to Reviewer #1:

The study by Luján et al provides evidence that endogenous cannabinoid signaling in dopamine neurons contributes to invigoration of behavior by dopamine and encoding of reward-predictive cues by dopamine transients. The study has many strengths, including technical expertise and converging approaches, inclusion of males and females, the use of conditional knock-outs, inclusion of df and other statistical information, and consideration of a range of behaviors when phenotyping mice. The manuscript is well written, which is no small feat for such a short paper.

1. My major concern/question is about the control groups used to compare the conditional knockout. Specifically, most of the experiments compare DGLa-f/f mice with the conditional KO to WT C57BL6/J mice rather than to DGLa-f/f mice without the conditional KO (or SHAM). The authors first cross DAT-cre mice with WT C57, DGLa-f/-, and DGLa-f/f, and find that the resulting lines do not differ in a variety of behaviors, but do differ in motor coordination (roto-rod). It is possible that reduced motor coordination could impact the speed of operant responding – how do the inter-press intervals compare across groups? The remainder of the study used DGLa-f/f mice with a conditional KO using AAV2/10-TH-cre (or other AAV) in the VTA. The one test I found of the DGLa-f/f SHAM mice in effortful responding was in Supplemental Figure 4B – these data showed that the number of presses in a PR task was similar in the DGLa-f/f SHAM versus the C57 WT, and the conditional KO reduced presses. Adding a comparison of the speed of lever press behavior (inter-press interval) across the three groups would inform the interpretation of the PR data – can the rats press quickly, or do they have a motor deficit (e.g. motor incoordination)?

We thank the Reviewer for their kind words and for their careful consideration of our experimental controls. Regarding the first point of the review, we would like to stress that *no reduced motor coordination* was observed in DGLa^{f/f} animals. Indeed, Extended Data Fig. 1 (now excluded from the Manuscript as per the other reviewers' suggestions) showed that DGLa^{f/f} displayed *improved* motor coordination compared to WT mice in the rotarod test (something that Reviewer #2 also pointed out). In addition, in conversation with Dr. Arlt, our reviewing editor, we considered carefully how to address this criticism and she agrees that the findings reported in the rotarod figure clearly agree with our justification for a lack of motor effects as the cause of the observed changes in behavior. In any case, we agree with the reviewer that examination of inter-response intervals during PR responding is thoughtful, and we think it strengthens the notion that no extraneous genotypic effects were present in the reported findings. In particular, we have added a new series of figures and analyses (Fig. 2) based on the data obtained with the animals tested on PR and previously highlighted on Extended Data Fig. 4. As the reviewer pointed out, this run included WT as well as DGLa^{f/f} sham and DGLaTH cKO mice, allowing us to properly control for the differences between WT C57BL6/J and DGLa^{f/f} animals. In short, no differences between groups were found in terms of FR1 responding or PR inter-press intervals (Fig. 2f).

Next, we explored how individual lever presses spontaneously clustered into 'bursting' events during the PR session. The rationale for this approach is that motor coordination impairments should be reflected in measures related to the individual responses within each lever pressing burst. However, measures related to the bursts themselves, i.e., number of response bursts/session, should be independent of motor execution and reflect task engagement (a motivational metric) instead. Again, the observed data support the lack of non-specific

differences, indicating that response speed, frequency and density did not vary across groups (Fig. 2h,i). Instead, the reduction of PR breaking points was mirrored by a reduction in the number of times DGL α^{TH} cKO mice engaged in the task (number of response ‘bursts’) (Fig. 2k). More importantly, we also explored the distributions of all these metrics to discard differences at high-frequency responding, where motor impairments would be more evident. Our data also discards total locomotor output alterations even at high-responding frequencies (Fig. 2g,h).

In short, all experimental mice display high frequencies of lever pressing behavior (including DGL α^{TH} cKO) and the reported decrease in PR breaking points is due to a reduction in the number of times animals engaged in clustered bursts of reward-oriented behavior.

The remainder of the comparisons compared C57 WT to the DGL α -f/f conditional KO. In the studies assessing the use of predictive cues to guide behavior (VTO), C57BL6/J mice are used as a control rather than DGL α -f/f SHAM mice. But in order to interpret the behavioral deficit, we need to know how the DGL α -f/f SHAM perform in the VTO task. This is important because the VTO task includes higher order conditioning that is more than simply effortful (such as a FR1 or PR task) – it should be clear that the DGL α -f/f SHAM mice perform as well as C57 mice in order to interpret the deficits observed in the conditional KO mice.

The reviewer highlights another point related to our experimental controls with room for improvement and we agree that VTO performance is fundamentally different to that of PR. Indeed, a DGL $\alpha^{\text{f/f}}$ *sham* group underwent VTO testing along their WT and DGL α^{TH} cKO counterparts. For internal control purposes, a set of *sham* animals is always tested to discard potential non-specific effects. However, this time DGL $\alpha^{\text{f/f}}$ *sham* mice were excluded from further analyses because they were not recorded with FSCV (Fig. 4). Nevertheless, we have now added this group to the VTO behavioral findings and this is now depicted in Extended Data Fig. 6. Our behavioral metrics indicate that DGL $\alpha^{\text{f/f}}$ *sham* are comparable to WT mice in terms of conditioned response latencies, number of rewards earned, omission errors and inactive lever presses during this particular task.

2. This concern carries into Figure 3, which shows that the dopamine release associated with 4 sessions of VTO differs between C57 WT to the DGL α -f/f conditional KO. The two lines of mice exhibited different dopamine release amplitude to the cue and reward from the first session, which persisted across 4 sessions. As differences in cue-evoked versus reward-evoked dopamine have been reported in various mouse and rat strains (e.g. with sign-tracking and goal-tracking), it is important to know how much of the difference in dopamine from the C57 mice is due to the conditional KO and how much is due to the DGL α -f/f mice. Note that there are two cues in this task – the light/tone cue and the lever extension, and in the representative trace the DGL α -f/f conditional KO mouse shows dopamine release to the lever extension, but not to the light/tone cue. Separating the dopamine response to these cues may uncover starker effects of the conditional KO and help to interpret the precise difference in dopamine signaling.

We appreciate the reviewer’s suggestion to further analyze these signals around the lever extension event, as this is obviously of interest due to its salient properties. To compare the dopamine response following lever extension, signals were analyzed on a trial-by-trial basis

during the period between lever extension and lever press, or over a 5-second window, if no lever press occurred during this period. Because the length of the signal varied across trials and between animals, we analyzed the maximal change (peak height) rather than area under the curve. We found a significant effect of time (i.e., the signal increased across sessions), but no significant difference between groups or its interaction. This analysis is now included in revised Figure 4, and a graphical depiction of the event-centered signals is also included in Supplemental Figure 6. Further, we have included our interpretation of the change in lever-evoked dopamine release in the Discussion (shown below).

P.12: lever extension (a proximal predictor of reward availability) triggered normal NAc dopamine elevations known to precede action initiation toward appetitive stimuli (Syed et al., 2016 Phillips et al., 2003; Owesson-White et al., 2008). The increased lever-evoked dopamine signal across sessions may be indicative of faster action initiation toward the lever (i.e., decreased latency to press), which was evident in both groups (though at different rates). However, this signal *was not transferred to the antecedent cue* following deletion of DGL α in VTA dopamine neurons.

3. Similarly, the behavioral response to the lever is described as latency to approach/press, but what is the behavioral response to the light/tone in the two genotypes?

Unfortunately, we are unable to calculate the behavioral response to the light/tone because we do not have video recordings of each session. However, and in accordance with our prior work (Owesson-White 2008; Oleson 2012), the decrease in latency to lever press across trials indicates that animals learned that the light/tone signaled forthcoming lever extension and the ability to earn a sucrose reward.

Additional comments:

4. While the paper states that both males and females were used, no indication of the proportion of each sex is provided. Please add #M and #F per group in the figure legends.

Male and females have now been distinguished in each figure (Males: closed circle, Females: open circle) and statistically compared for each task. A significant sex difference was only observed for body weight, and this is now highlighted in Figure 1 legend.

5. The paper states that mice between 3-6 months old were used. As that's a broad range, and older rodents can exhibit lower dopamine release and take longer to learn tasks, please show that mouse age is generally matched across groups/genotypes.

The reviewer is right in their appreciation that 3 months is a broad range of ages for mice. We ran the suggested analysis and confirmed that there was no significant difference in age between groups ($t_{16} = 0.492$, $p = 0.63$) that could differentially affect dopamine release or learning rates.

6. It would be helpful to the reader to refer to the conditional KO mice as DGL α -f/f cKO rather than just DGL α -f/f (which could be SHAM, non-manipulated, or cKO).

Following the reviewer's suggestion, we have relabeled our groups throughout the manuscript and figures to improve clarity. In the revised version of the manuscript, KO mice are now

referred to as $DGL\alpha^{TH}$ cKO to better highlight that $DGL\alpha$ deletion was limited to TH + neurons. More importantly, non-manipulated $DGL\alpha^{f/f}$ mice are now referred to as $DGL\alpha^{f/f}$ *sham*.

7. Please add to the methods or figure caption how the change in dopamine concentration was measured – what was the baseline for each event? Did the cue response AUC include the lever extension?

Details have been added to the Methods section. Baseline was the 0.5 s preceding cue onset. The lever extension occurred 5 s later and was not included in the analysis of the cue response.

Answers to Reviewer #2:

In the study by Lujan et al, the authors examined the role of endogenous 2-AG mobilization in the ventral tegmental area (VTA) on behavior and dopamine release in the nucleus accumbens (NAc). To address this question, they used a viral-genetic approach to reduce the expression of diacylglycerol lipase a (DGLa), which is critical for 2-AG synthesis. Given that many studies on cannabinoid function are based on agonist administration, this approach and the resulting findings are highly significant to the field as they can uncover the endogenous function of endocannabinoid signaling on neurotransmission and behavior. Overall, this study indicates that the dopamine response to visual cues and the corresponding behavioral response to those cues are regulated by endogenous 2-AG signaling in the VTA. A particular strength of the manuscript is the robust behavioral effects of knocking down DGL expression in the VTA on cued (but not uncued) tasks. While there is significant enthusiasm for this study, there are some notable issues that need to be addressed that are listed below. Many of the concerns can likely be addressed if this paper were reformatted for a full-sized article versus the current brief communication format.

Major concerns (in order of appearance):

1. Please provide the behavioral performance of the animals on the FR1 training prior to the PR sessions. Also, clarify the session duration of the FR1 training. Specifically, one needs to know if decreasing DGL expression in the VTA led to differences in the operant responding during the FR1 phase, as this could confound the interpretation of the data from the subsequent PR sessions. Please also provide further details on how many PR sessions the animals underwent.

The reviewer is right in their consideration that more details are needed regarding the training that each batch of mice was subjected to. This and other details are now clarified in the 'Behavior' section of the methods, under the 'Operant training' and 'Fixed ratio 1' subsections (page 23) of the revised manuscript. It is now stated that animals underwent four PR sessions and that FR1 training sessions lasted 30-min. Figure panels have been re-arranged to better represent the training conducted by mice of each experiment.

Regarding the concern of the reviewer (pre-existing changes in FR1 that would later impact PR performance), we have added a new figure (Figure 2) that should address it. Specifically, we do not detect changes in sucrose operant responding during the FR1 phase (Fig. 2a,b) in the cohort of WT, DGL α^{ff} sham, DGL α^{TH} cKO mice that were later tested on PR (Fig. 2c,d). This data is consistent with results obtained from the oICSS cohort of mice, showing no training or responding differences during FR1 (Fig. 3l,m).

2. The behavioral experiments provided in Extended Data Fig 1 using the DAT cre x DGL f/f mice are not an appropriate control for the experiments in the rest of the study, which instead used a viral approach to express Cre in TH expressing neurons in the VTA of adult DGL f/f mice. The solely genetic approach used in Extended Data Fig 1 could lead to developmental compensation. The behavioral characterization experiments examining if there are gross impairments should use the same viral-genetic approach that is used in the main figures. Additionally, the rationale for these chosen behavioral assays in Extended Data Fig 1 wasn't evident. Specifically, what is the purpose of the Y-maze assay? And provided the results are consistent when using the viral-genetic approach, how should one interpret the finding that knocking down DGL expression in the VTA improves motor skills on the rotarod assay? I am

also not sure these experiments are needed given the data shown in Fig 1J-L. More important control experiments are outlined in Point 7 below.

Our initial purpose with Extended Data Fig. 1 approach (DATcre x DGLf/f) was to rule out extraneous phenotypic alterations due to the conditional removal of DGL α in a *worst-case scenario* (a constitutive deletion). That is, if no gross impairments are evident in animals lacking DGL α in all dopamine neurons from development then fewer deficits should emerge in VTA-DGL α^{TH} cKO mice transduced during adulthood. However, we agree with the reviewer that this approach could not rule out developmental compensation effects and that the purpose of some of the tests employed is beyond the scope of the current study. We have removed DATcre x DGLf/f experiments and Extended Data Fig. 1 from the revised version of the manuscript and focused on the suggested experimental controls.

3. Please report the sex for the subjects in each figure. Also please show the individual data points in the summary bar graphs throughout the study.

We have modified all figures and captions to show desegregated numbers pertaining to each sex. In all figures, open circles now represent data from female subjects and closed circles from male animals. In addition, we have expanded the “Subjects” section, in Methods, to describe how sex differences were analyzed. In summary, no sex differences, or sex interactions, were found in any of the measures examined with the exemption of bodyweights (Fig. 1i). Lower female bodyweights were equally reported among WT and DGL α^{TH} cKO genotypes.

4. The absence of DSE in the DGLf/f mice is strong electrophysiological evidence that the viral-genetic approach is working as intended. However, it is not clear that this electrophysiological effect is contributing to the observed effects on dopamine release and behavior. This again may be an issue with the brief communication format. When reformatting this paper, it would be recommended to highlight this data as functional evidence that the manipulations are working (versus saying this is a plausible mechanism underlying the behavioral effects).

We thank the reviewer for their suggestion, and we have now stressed that the absence of DSE in DGL α^{TH} cKO mice is strong electrophysiological evidence that the deletion is working as intended, but that it should not be interpreted as the mechanism for the behavioral and dopaminergic effects reported. This is thoroughly discussed in page 11 (Discussion section):

“Of note, the loss of DSE reported in DGL α^{TH} cKO animals is not thought to directly participate in the behavioral and dopaminergic effects reported here. Blunting CB1R-induced inhibition of glutamatergic influences (DSE) should result in the facilitation of dopamine cell function and its related functions, in contrast to the observed phenotype. Instead, we postulate that a loss of DSI was responsible for the observed effects, as it is consistent with a disinhibition of GABAergic input onto VTA dopamine neurons”.

As described below (Point 13), the manuscript has been re-formatted to a full-article type of publication.

5. There are some questions regarding the analysis of the dopamine signals. It isn't clear why the cue response is calculated over a 3s window when the cue is presented for 5s. I am assuming that the 3s cue window is designed to match the 3s reward window for the analysis

in Fig. 3G. But the analysis in Fig. 3G is also not intuitive regarding how it was calculated. Specifically, what is contributing to the denominator? If it is just the sum of the calculated Cue response (over 3 s) and the calculated reward response (over 3 s), then this means there is a 2 s window during the cue that is not being analyzed? Further details are needed regarding this analysis. In short, I would recommend performing the cue analysis over the full 5s of the cue.

We apologize for the mistake. A 5 second window was indeed used to calculate both the cue- and reward-evoked dopamine response. We have modified every mention throughout the manuscript to report the correct window analyses (5s) employed for the cue- and reward-evoked dopamine traces.

6. The dopamine analysis is focused on the presentation of the cue lights and following the reward delivery. However, the average dopamine traces illustrate the dopamine response also increase when the lever is presented, though this is not quantified. The lever-evoked response is important to examine as this response seems to increase over training in the mice that lack DGL in the VTA (Fig. 3H). Conceptually, the lever presentation could be seen as another reward-predictive cue, so one should be cautious on how to interpret these results. In short, it is important to quantify the lever-evoked dopamine response. I would recommend quantifying the average dopamine between the lever presentation and the delivery of the food reward, which would have to be calculated on a trial-by-trial basis.

We appreciate the comment and agree this is an important analysis. We have provided a detailed description throughout the revised version of the manuscript. Specifically, the results of the lever-evoked dopamine response can be found in Fig. 4G. The entire period of time included between lever extension and lever press was used to characterize this response on a trial-by-trial basis.

As suggested by the reviewer, lever-evoked dopamine responses were similar in size and rate of change through training between groups, suggesting that removal of DGL from VTA dopamine neurons did not abolish *all* outcome-predictive encoding but only that of *the most distal* predictive cues (in this case, the compound cue, light+sound). This is a crucial finding and we have conceptually re-framed our interpretation of the results. In particular, we now argue that predictive encoding is still preserved in DGL α^{TH} cKO mice but its ability to progressively transfer incentive salience to the earliest predictor (compound cue) of forthcoming reward is abolished. This could be related to diminished cue-elicited reward seeking and unexpected presentation of reward, consistent with the greater reward-evoked dopamine release of DGL α^{TH} cKO animals, and future experiments from our laboratory will formally test this hypothesis.

7. One of the most intriguing findings was the absence of the dopamine response to the cue light turning on in the VTO task in the mice lacking DGL in the VTA (Fig. 3H). However, as discussed above, these mice seem to exhibit an increase in the dopamine response to the lever presentation, which is the most proximal reward-predictive stimulus. This opens the question if the absence of DGL in the VTA is impairing the detection of visual stimuli or alternatively is impairing the detection of distal reward-predictive cues. To address this question one could train mice on a Pavlovian conditioning task using visual stimuli or audio stimuli (in separate groups of animals). If there is a light above the food tray, one could illuminate that light for 5s before the reward delivery. Alternatively, one could illuminate both

of the cue lights (no lever extension) for 5s before the reward delivery. For the audio Pavlovian task, one could present a tone for the 5s before the reward delivery. These stimuli should presumably promote goal tracking behavior since no sign tracking behavior can be exhibited. By comparing the anticipatory head entries during the cues between WT and DGLf/f mice in the audio and visual Pavlovian tasks, one can determine if the deficit in DGLf/f mice is to visual stimuli (if impairments are seen only in the visual Pavlovian task) or to distal reward-predictive cues (if no impairments are seen in both Pavlovian tasks) or is evident only in operant tasks (if impairments are seen in both Pavlovian tasks).

We appreciate the reviewer's creative suggestion and apologize for not including this in the methods, but a compound cue (cue light + tone) was used to indicate trial onset. Our previously published work has successfully made use of this compound light+tone stimulus as the standard procedure (Lujan et al., 2023; Covey et al., 2021). Thus, visual impairment cannot explain diminished cue-evoked signal in the DGL α^{TH} cKO mice. The methods section has been modified to properly describe the stimulus employed (subsection 'Apparatus', page 23).

I hesitate to ask for new experiments, but in this case I feel that it is quite important given the stated conclusions of the study. However, if there is no effect of training on the lever-evoked dopamine response in the DGLf/f mice (see Point 6), the proposed Pavlovian experiments would be unnecessary as the data would then indicate that training did not impact the dopamine response to either the distal cue (light) or the proximal cue (lever).

Lever-evoked dopamine response analyses indicate that dopamine elevations to the proximal cue increased across training sessions (two-way RM ANOVA; $^{\text{session}}F_{3,48} = 4.02, p = 0.01$) in a similar way for the two groups ($^{\text{genotype}}F_{1,16} = 0.23, p = 0.63$; $^{\text{interaction}}F_{3,48} = 0.34, p = 0.78$). This indicates that DGL α^{TH} cKO mice can still form associations with the most proximal cue (lever extension) available in the VTO task, as shown by the qualitative similarities of the voltammetry signals at the lever presentation seen in both groups. However, and more importantly, transference of salience to the most distal cue (the earliest predictor of reward) was specifically abolished following DGL α removal from VTA dopamine neurons. As mentioned above (Point 6), we have re-framed our interpretations and even updated the manuscript's title to reflect this change. Regarding the sensory perception-related concerns raised by the reviewer, we believe that no fundamental differences exist between the distal and proximal cue, as both the compound cue and the lever extension share sensory modalities (visual and auditory).

8. Histology figures for voltammetry electrodes and optogenetic fibers are missing from the manuscript.

We and others have historically had trouble histologically verifying the voltammetry electrode placement due to their size (5 μm diameter and 75 μm length) and the fact that they are made out of carbon. However, the ability to detect dopamine indicates that sensors were implanted in the nucleus accumbens. Moreover, the ability to drive intra-cranial self-stimulation with the optogenetic activation, which requires dopamine release in the NAc (Steinberg et al., 2014), and simultaneously record dopamine release evoked by the optical stimulation provides strong functional evidence that we accurately targeted dopamine projections to the NAc.

Text issues:

9. Details on the sucrose consumption and OFT refer to Q175 mice, which presumably was data contained in a different study.

We apologize for the mistake and thank the reviewer for their helpful editing suggestions. Mentions to Q175 mice have been removed from the text.

10. Figure 1D legend has the orientation backwards regarding the left/right of where the injections were given.

Lef/right orientations have been fixed in Fig. 1 legend.

11. Details on the sucrose feeding test are not provided (Fig. 1J). Only those for the 24hr sucrose preference test are provided (Extended Fig. 1). Or is the data in Fig. 1J reflecting the sucrose preference test but then converted to grams?

Details have been added to the Methods section. Mice were placed in an open-field arena (1m x 1m) and allowed free access to sucrose pellets for 30 minutes.

12. Text refers to AM experiments as Ext Fig 4, but it is Ext Fig 5. The dotted lines in the figure are confusing. Were those cells that underwent 2 rounds of DSE induction? If so, those should be removed from the analysis.

We agree with the Reviewer that the dotted lines appearing in Extended Data Fig. 4 were confusing. We have confirmed that these appeared due to an error on the graphing software and have been accordingly removed. To be clear, no cells underwent two rounds of DSE induction.

13. This should be written and formatted as a full manuscript.

To accommodate the Reviewer's suggestion, new figures and expanded discussion, we have re-format the Manuscript as a full-article type of publication. This suggestion has greatly improved the discussion and interpretation of the obtained results.

REVIEWERS' COMMENTS

Reviewer #1 (Remarks to the Author):

The authors have sufficiently responded to my concerns.

Reviewer #2 (Remarks to the Author):

The authors were very responsive to the prior round of critiques. This is an excellent manuscript, especially now with its formatting as a regular manuscript. The only minor remaining issue is that the individual data points are not presented in Fig. 3. Great work.

Answers to Reviewer #1:

The authors have sufficiently responded to my concerns.

We thank the Reviewer for their devoted efforts improving the quality of this manuscript.

Answers to Reviewer #2:

The authors were very responsive to the prior round of critiques. This is an excellent manuscript, especially now with its formatting as a regular manuscript. The only minor remaining issue is that the individual data points are not presented in Fig. 3. Great work.

The individual data points are now presented with the panels graphed in Fig. 3.

We thank the Reviewer for their kind words and insightful comments and critiques.